# A Unifying View of Vector, Product and Scalar Quantization:
# An Information-Theoretic Perspective

## Abstract

Discrete visual tokenization, predominantly driven by vector, scalar, and product quantization, lacks a unifying conceptual framework that elucidates the impact and tradeoffs of different quantization optimization objectives. In this paper, we propose a unified information-theoretic framework to shed light on these considerations. To do so, we view quantization as information compression and define the information loss (quantization error), compression ratio, and input/output as information-theoretic quantities. Using this framework, we resolve three central open questions: First, we theoretically prove and empirically demonstrate that minimizing quantization error, rather than maximizing codebook utilization, is the paramount optimization objective for ensuring training stability and reconstruction fidelity. Second, we establish two critical fairness conditions for intrinsic algorithm comparison: controlling the latent feature distribution variance and ensuring identical compression ratios. Third, we demonstrate, both theoretically and empirically, that under these conditions, modern vector quantization outperforms scalar and product quantization at minimizing quantization error. Our work provides a foundational reframing of quantization algorithms, resolving conceptual ambiguities and providing the first artifact-free comparison that establishes quantization error minimization as the core optimization criterion.

## 1 Introduction

Discrete visual tokenization has achieved remarkable success in recent years, driven by the evolution of quantization algorithms, such as advances in vector quantization (VQ) (Esser et al., 2021; Sun et al., 2024; Tian et al., 2024), product quantization (PQ) (Jégou et al., 2011; Guo et al., 2024; Li et al., 2025), and scalar quantization (SQ) (Mentzer et al., 2024a; Yu et al., 2024; Zhao et al., 2025; Han et al., 2025). However, the community's conceptual understanding has plateaued. Notably, *full codebook utilization* continues to dominate as the primary optimization objective (Dhariwal et al., 2020; Lee et al., 2022; Zheng & Vedaldi, 2023; Zhu et al., 2024), while other essential algorithmic trade-offs receive limited attention. This gap underscores the urgent need for a unifying conceptual framework that comprehensively addresses the impact and tradeoffs of different quantization objectives.

In this paper, we propose a unified information-theoretic view of quantization to systematically address these considerations. By conceptualizing quantization algorithms as information compression systems, we formalize the information loss (quantization error), compression ratio, and input/output representations as fundamental information-theoretic quantities. These quantities can be rigorously defined based on the number of bits required to encode the underlying information. Building upon this foundation, this paper resolve three central open questions in the field.

First, we investigate the primary optimization objective for quantization algorithms. Grounded in principles from information compression systems, minimizing information loss (or quantization error)—rather than maximizing codebook utilization—yields superior training stability and reconstruction performance. We provide both theoretical and empirical evidence to solidify quantization error minimization as the paramount objective. Specifically, we theoretically prove that minimizing quantization error necessarily implies full codebook utilization, while the converse does not hold. Empirically, we demonstrate significantly stronger correlations between quantization error and reconstruction fidelity (measured by r-FID) compared to those observed with codebook utilization.

Second, we establish two necessary conditions for a fair comparison of the intrinsic effectiveness of quantization algorithms. The first condition expresses that latent feature distributions must be identical across all compared algorithms. This is because, under optimal codebook conditions, the quantization error scales linearly with the variance of latent feature distributions. When the quantization error is dominated by feature variance, the intrinsic effectiveness of quantization algorithms can be obscured, potentially leading to erroneous conclusions based on direct quantization error comparisons. The second condition expresses that compression ratios must be held constant across all algorithms by using identical token counts and codebook sizes. This is because both token counts and codebook sizes significantly influence quantization error. Only by strictly adhering to both conditions can we accurately compare the intrinsic effectiveness of different quantization algorithms.

Third, we examine the intrinsic effectiveness of quantization algorithms under rigorously controlled conditions. Since SQ, PQ, and VQ form a hierarchy in which VQ generalizes PQ and PQ in turn generalizes SQ, VQ methods inherently offer greater modeling flexibility and higher performance potential. However, early studies on VQ highlighted severe codebook collapse issues, particularly with large codebook sizes (Dhariwal et al., 2020; Takida et al., 2022; Yu et al., 2022; Lee et al., 2022; Zheng & Vedaldi, 2023), leading to poor performance compared to SQ and PQ (Mentzer et al., 2024a; Yu et al., 2024; Zhao et al., 2025; Guo et al., 2024). To resolve this conflict, through rigorous theoretical analysis, we re-establish that VQ algorithms intrinsically exhibit superior effectiveness compared to PQ and SQ algorithms. Our empirical study further demonstrates that advanced VQ algorithms (Zhu et al., 2024; Fang et al., 2025; Anonymous, 2025) experience smaller information loss and better reconstruction performance, under the two aforementioned fair-comparison conditions.

Our key contributions are as follows:

1. **A Theoretical and Conceptual Framework**: We introduce a unifying information-theoretic framework that conceptualizes quantization algorithms as information compression systems. Using this framework, we resolve three aforementioned central open questions.

2. **An Empirical Validation Under Fairness Constraints**: Under strict adherence to two fair conditions, our benchmark yields artifact-free evaluation under which: (i) quantization error constitutes a more consequential optimization objective than full codebook utilization, and (ii) VQ exhibits fundamental superiority over PQ/SQ baselines on this primary distortion metric. This methodology-first approach yields the first artifact-free comparison establishing quantization error as the paramount optimization criterion.

## 2 BACKGROUND

### 2.1 DISCRETE VISUAL TOKENIZER

Contemporary visual generative models primarily follow two paradigms (Wang et al., 2024): language model-based or diffusion-based approaches. The former leverages sequence modeling to formulate visual generation as next-token prediction (van den Oord et al., 2017; Esser et al., 2021; Sun et al., 2024), relying on quantization-based tokenizers such as VQVAE (van den Oord et al., 2017). Diffusion models (Ho et al., 2020; Song et al., 2021a;b), conversely, employ continuous tokenizers (e.g., VAEs (Kingma & Welling, 2014; Rombach et al., 2022)) to encode images into compact latent distributions. This work concentrates on the study of discrete visual tokenizers which deploy quantization techniques—specifically vector quantization (VQ), scalar quantization (SQ), and product quantization (PQ) strategies.

As depicted in Figure 1, the discrete visual tokenizer typically consists of three key components: an encoder $\mathcal{E}_\theta$, a quantization module $\mathcal{Q}_\phi$, and a decoder $\mathcal{D}_\varphi$. Given an input image $\boldsymbol{x} \in \mathbb{R}^{H \times W \times 3}$, the encoder $\mathcal{E}_\theta$ produces a set of $d$-dimensional feature embeddings

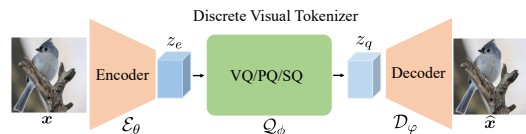

Figure 1: The illustration of discrete visual tokenizer.

$\boldsymbol{z}_e = \mathcal{E}_\theta(\boldsymbol{x}) \in \mathbb{R}^{(H/f) \times (W/f) \times d}$, with a spatial downsampling factor of $f \times f$. The quantization module then discretizes these continuous features, yielding a discrete token $r^{ij} \in \mathbb{N}$ and a quantized latent spatial features $\boldsymbol{z}_q^{ij} = \mathcal{Q}_\phi(\boldsymbol{z}_e^{ij})$. The discrete tokens $\{r^{ij}\}$ are used to train generative models, while the quantized latents $\{\boldsymbol{z}_q^{ij}\}$ are decoded to reconstruct the image $\widehat{\boldsymbol{x}} = \mathcal{D}_\varphi(\boldsymbol{z}_q)$.

## 2.2 Vector Quantization

Vector quantization (VQ) provides an early approach for learning discrete visual tokenizers (van den Oord et al., 2017). This method employs a learnable codebook $\phi = \{\mathbf{e}_k\}_{k=1}^K \subset \mathbb{R}^d$ containing $K$ code vectors, where $d$ is the vector dimension. The VQ module discretizes continuous spatial features $\mathbf{z}_e^{ij}$ by assigning it to its nearest codebook entry:

$$k^* = \underset{k \in \{1,2,...,K\}}{\arg\min} \|\mathbf{z}_e^{ij} - \mathbf{e}_k\|_2^2, \tag{1}$$

yielding a discrete visual token $r^{ij} = k^* \in \{1, 2, ..., K\}$. The quantized latent representation is then given by:

$$\mathbf{z}_q^{ij} = \mathcal{Q}_\phi(\mathbf{z}_e^{ij}) = \mathbf{e}_{r^{ij}}, \tag{2}$$

where $\mathcal{Q}_\phi$ denotes the quantization operator parameterized by codebook vectors $\phi$.

Early VQ algorithms commonly suffer from severe codebook collapse (Dhariwal et al., 2020), where only a sparse subset of code vectors receive meaningful gradient updates, leaving the majority of embeddings underutilized (Dhariwal et al., 2020; Takida et al., 2022; Yu et al., 2022; Lee et al., 2022; Zheng & Vedaldi, 2023). This issue is particularly pronounced at large codebook sizes $K$ (Zheng & Vedaldi, 2023; Mentzer et al., 2024b). While substantial research has developed improved VQ learning strategies (Zhu et al., 2024; Dhariwal et al., 2020; Williams et al., 2020; Razavi et al., 2019; Zheng & Vedaldi, 2023; Zhang et al., 2023; Ramesh et al., 2021) and some effectively mitigate codebook collapse (Zhu et al., 2024; Fang et al., 2025; Anonymous, 2025), we contend that minimizing quantization error constitutes a more critical optimization objective than maximizing codebook utilization.

## 2.3 Product Quantization

Product quantization (PQ) constitutes an alternative quantization framework interpretable as an ensemble of VQ modules (Jégou et al., 2011; Guo et al., 2024; Li et al., 2025). Specifically, PQ partitions continuous spatial features $\mathbf{z}_e^{ij} \in \mathbb{R}^d$ into $M$ distinct subvectors:

$$\mathbf{z}_e^{ij} = \bigoplus_{m=1}^M \mathbf{z}_m^{ij}, \quad \mathbf{z}_m^{ij} \in \mathbb{R}^{d_m}, \text{where } \sum_{m=1}^M d_m = d. \tag{3}$$

Here, $\bigoplus$ denotes channel-wise concatenation. Each subvector $\mathbf{z}_m^{ij}$ is quantized via an independent VQ module $\mathcal{Q}_{\phi_m}$:

$$\text{Quantized feature:} \quad \widehat{\mathbf{z}}_m^{ij} = \mathcal{Q}_{\phi_m}(\mathbf{z}_m^{ij}); \quad \text{Discrete token:} \quad r_m^{ij} \in \{1, \ldots, n_m\}. \tag{4}$$

The composite quantized vector and its corresponding token are given by:

$$\mathbf{z}_q^{ij} = \bigoplus_{m=1}^M \widehat{\mathbf{z}}_m^{ij}, \quad r^{ij} = r_1^{ij} + \sum_{m=2}^M \big(\prod_{k=1}^{m-1} n_k\big) r_m^{ij}. \tag{5}$$

Each subcodebook $\phi_m = \{\mathbf{e}_{m,k}\}_{k=1}^{n_m}$ contains $n_m$ embeddings, inducing an implicit global codebook size $K = \prod_{m=1}^M n_m$. This subspace decomposition mitigates codebook collapse (Guo et al., 2024). Crucially, while supporting $K$ distinct codewords, PQ requires optimization of only $\sum_{m=1}^M n_m$—reducing training complexity substantially. The overall PQ process simplifies to:

$$\mathbf{z}_q^{ij} = \mathcal{Q}_\phi(\mathbf{z}_e^{ij}) \quad \text{with} \quad \phi = \{\phi_m\}_{m=1}^M; \quad \mathcal{Q}_\phi = \bigoplus_{m=1}^M \mathcal{Q}_{\phi_m}. \tag{6}$$

## 2.4 Scalar Quantization

Scalar quantization (SQ) discretizes continuous scalars, representing an extreme case of product quantization (PQ) with $M = d$. Continuous spatial features $\mathbf{z}_e^{ij} \in \mathbb{R}^d$ decompose dimension-wise into $d$ scalar components:

$$\mathbf{z}_e^{ij} = \bigoplus_{m=1}^d z_m^{ij}, \quad \text{where} \quad z_m^{ij} \in \mathbb{R}. \tag{7}$$

Independent scalar quantizers $\mathcal{Q}_{\phi_m}$ operate per dimension:

$$\widehat{z}_m^{ij} = \mathcal{Q}_{\phi_m}(z_m^{ij}); \quad r_m^{ij} \in \{1, \ldots, n_m\}, \tag{8}$$

yielding quantized features and discrete tokens. The full quantized vector and composite token index are constructed as:

$$z_q^{ij} = \bigoplus_{m=1}^{d} \widehat{z}_m^{ij}, \quad r^{ij} = r_1^{ij} + \sum_{m=2}^{d} \Big( \prod_{k=1}^{m-1} n_k \Big) r_m^{ij}. \tag{9}$$

Each subcodebook $\phi_m = \{e_{m,k}\}_{k=1}^{n_m}$ contains $n_m$ discrete scalars, yielding a global codebook size $K = \prod_{m=1}^{d} n_m$ while maintaining optimization complexity $\mathcal{O}(\sum_{m=1}^{d} n_m)$. This formulation is functionally equivalent to PQ (Equation (6)). Notably, FSQ (Mentzer et al., 2024a) shares identical subcodebooks across dimensions, that is, $\forall i \neq j, \phi_i = \phi_j$ with codewords constrained to finite integers. LFQ (Yu et al., 2024) employs binary quantization with $\phi_m = \{-1, 1\}$. BSQ (Zhao et al., 2025) projects features onto the unit sphere pre-quantization, yielding normalized codebooks $\phi_m = \{-\frac{1}{\sqrt{d}}, \frac{1}{\sqrt{d}}\}$.

## 3 ON THE INTRINSIC EFFECTIVENESS OF QUANTIZATION ALGORITHMS

**Challenges.** Discrete visual tokenizers typically implement a cascaded two-stage compression pipeline: First, an encoder-decoder transforms raw visual signals into continuous latent representations, which are then discretized into tokens via quantization algorithms (e.g., VQ, PQ, or SQ). However, this cascaded architecture fundamentally obstructs rigorous assessment of quantization algorithms' intrinsic effectiveness. Empirical evaluations based on this framework often yield misleading conclusions due to inconsistent experimental baselines. A contributing factor to this limitation is the substantial computational overhead involved in end-to-end tokenizer training. As a result, many studies resort to leveraging pre-existing experimental results, which may inadvertently introduce confounding variables at the encoder-decoder stage (Zhu et al., 2024; Li et al., 2025; Ma et al., 2025). These methodological inconsistencies lead to unfair comparisons between quantization approaches, ultimately producing unreliable conclusions about their intrinsic capabilities.

Specifically, as categorized in Table 1, these sources of unfairness primarily manifest through disparities in: (i) model parameters, (ii) architectural designs, (iii) discriminator configurations, (iv) training datasets, (v) training epochs, and (vi) computational resources. Early VQ-based tokenizer studies often employed encoder-decoder architectures with constrained model capacity (e.g., CNN-based U-Nets (Ronneberger et al., 2015)), trained with limited computational budgets on smaller-scale datasets like ImageNet-1k (Deng et al., 2009) and paired with low-capacity discriminators (e.g., PatchGAN (Isola et al., 2017)). In contrast, modern discrete tokenizers typically utilize significantly larger-scale encoder-decoder structures (e.g., transformer-based SEED (Ge et al., 2023)), leverage expanded datasets such as OpenImages (Kuznetsova et al., 2018), and employ high-capacity discriminators (e.g., StyleGAN (Karras et al., 2019)) while consuming substantially greater computational resources—all contributing to improved reconstruction fidelity. However, these methodological discrepancies obscure the intrinsic effectiveness of quantization algorithms, as performance gains attributable to algorithmic advances become conflated with improvements from enhanced architectural capacities and training resources.

**Solutions.** To investigate the intrinsic effectiveness of quantization algorithms, we isolate their core contribution by controlling for the aforementioned confounding factors. We introduce a unified information-theoretic framework that conceptualizes quantization as information compression. Using this framework, we address three central open questions, enabling rigorous theoretical and empirical comparison of intrinsic algorithm effectiveness.

Table 1: Comparison of tokenizer implementations: six key confounding factors. '-' indicates the factor is not provided, and "Para" denotes the parameter of the encoder-decoder architecture.

| Tokenizers | Para | Architectures | Discriminator | Training Datasets | Training Epochs | Training GPU Hours |
|---|---|---|---|---|---|---|
| VQGAN (Esser et al., 2021) | 68M | CNN U-Net | PatchGAN | ImageNet-1k | - | - |
| RQVAE (Lee et al., 2022) | 95M | CNN U-Net | PatchGAN | ImageNet-1k | 50 | - |
| VQGAN-LC (Zhu et al., 2024) | 68M | CNN U-Net | PatchGAN | ImageNet-1k | 20 | 32 × V100 - Hours |
| Llama GEN (Sun et al., 2024) | 68M | CNN U-Net | PatchGAN | ImageNet-1k | 40 | 2 × A100 200 Hours |
| VAR (Tian et al., 2024) | 104M | CNN U-Net | StyleGAN | OpenImages | 16 | 16 × A100 60 Hours |
| ImageFolder (Li et al., 2025) | - | Transformer SEED | StyleGAN | ImageNet-1k | 200 | 32 × A100 40 Hours |
| UniTok (Ma et al., 2025) | - | Transformer SEED | StyleGAN | OpenImages | - | 256 × A100 50 Hours |

# 4 AN INFORMATION-THEORETIC PERSPECTIVE

In this section, we present a unifying information-theoretic framework that models quantization algorithms as information compression systems. Utilizing this framework, we resolve three central open questions, addressed in Section 4.2, Section 4.3, and Section 4.4, respectively.

## 4.1 INFORMATION-THEORETIC QUANTITIES

We define the following core information-theoretic quantities: input information quantity, output information quantity, compression ratio, and information loss (quantization error). In Appendix J, we introduce our definitions and highlight their similarities and differences with Shannon entropy.

**Definition 1.** *Given the input $z_e \in \mathbb{R}^{h \times w \times d}$ to the compression system (specifically, the latent feature embeddings described in Section 2.1), the input information quantity $\mathcal{Q}_i$ is the amount of information (in bits) required to represent $z_e$. This is calculated as:*

$$\mathcal{Q}_i = h \times w \times d \times 32,$$

*where $h = H/f$ and $w = W/f$ denote the height and width of the latent features, respectively, $d$ is the channel dimension, and scalar values in $z_e$ are represented using the 32-bit floating-point format.*[1]

**Definition 2.** *Given the output $\{r^{ij}\} \in \mathbb{N}^{h \times w}$ of the compression system (specifically, the discrete visual tokens described in Section 2.1), the output information quantity $\mathcal{Q}_o$ is the amount of information (in bits) required to represent $\{r^{ij}\}$. This is calculated as:*

$$\mathcal{Q}_o = h \times w \times \log_2 K,$$

*where $K$ is the global codebook size, bounded by $K$, and $\log_2 K$ represents the maximum information (in bits) per token.*

**Definition 3.** *The compression ratio $\mathcal{Q}_r$ quantifies the reduction in information between the input and output of the compression system. It is defined as:*

$$\mathcal{Q}_r = \mathcal{Q}_i/\mathcal{Q}_o,$$

*where $\mathcal{Q}_i$ is the input information quantity and $\mathcal{Q}_o$ is the output information quantity.*

**Definition 4.** *The information loss $\mathcal{E}$, also referred to as the quantization error, measures the fidelity loss incurred during quantization. Given the input $z_e$ and the quantized latent spatial features $z_q$ (resulting from the compression process), it is defined as the squared Euclidean distance:*

$$\mathcal{E} = \|z_e - z_q\|_2^2.$$

To provide some intuition for these definitions, we provide concrete examples. For simplicity, we assume each subcodebook has identical size in both PQ and SQ, denoted as $K_1$ for PQ and $K_2$ for SQ. As shown in Table 2, we can precisely calculate the three information-theoretic quantities for VQ, PQ, and SQ: input/output information quantity, and compression ratio. For Residual Quantization (RQ) (Lee et al., 2022) and VAR quantization (Tian et al., 2024), these methods increase output information quantity by utilizing additional tokens, thereby achieving lower information loss. Notably, $\alpha_1$ denotes the number of residual quantization steps, while $\alpha_2$ represents the ratio of total multi-scale tokens in the VAR structure to the base spatial dimension ($h \times w$). For the ImageNet-256 benchmark, $\alpha_2 = \frac{680}{256} \approx 2.66$ when the spatial downsampling factor is 16 (Tian et al., 2024).

Table 2: Concrete examples for understanding definitions.

|  | VQ | PQ | SQ | RQ | VAR |
|---|---|---|---|---|---|
| Codebook Size | $K$ | $K_1^M$ | $K_2^d$ | $K$ | $K$ |
| Tokens | $h \times w$ | $h \times w$ | $h \times w$ | $h \times w \times (\alpha_1 + 1)$ | $h \times w \times \alpha_2$ |
| $\mathcal{Q}_i$ | $h \times w \times d \times 32$ | $h \times w \times d \times 32$ | $h \times w \times d \times 32$ | $h \times w \times d \times 32$ | $h \times w \times d \times 32$ |
| $\mathcal{Q}_o$ | $h \times w \times \log_2 K$ | $h \times w \times M \times \log_2 K_1$ | $h \times w \times d \times \log_2 K_2$ | $h \times w \times (\alpha_1 + 1) \times \log_2 K$ | $h \times w \times \alpha_2 \times \log_2 K$ |
| $\mathcal{Q}_r$ | $\frac{d \times 32}{\log_2 K}$ | $\frac{d \times 32}{M \times \log_2 K_1}$ | $\frac{32}{\log_2 K_2}$ | $\frac{d \times 32}{(\alpha_1 + 1) \times \log_2 K}$ | $\frac{d \times 32}{\alpha_2 \times \log_2 K}$ |

Based on the information-theoretic quantities defined previously, we derive an important observation: doubling the count of tokens $T$ corresponds to a *squared* increase in the required codebook size $K$.

---

[1]$z_e$ consists of scalar values represented using the 32-bit floating-point format.

This equivalence is formalized below:

**Finding 1.** *Equivalence between token count and codebook size.*
*The information-theoretic relationship between token count $T$ and codebook size $K$ is given by:*

$$\mathcal{Q}_o = 2 \times T \times \log_2 K = T \times \log_2(K^2).$$

This relationship implies that doubling $T$ is equivalent to squaring $K$ in terms of information capacity. Consequently, token count $T$ has a stronger influence than codebook size $K$ on the output information quantity (and potentially on information loss).

## 4.2 INFORMATION LOSS MINIMIZATION: THE PRIMARY OPTIMIZATION OBJECTIVE

Most existing quantization methods primarily address codebook collapse in compression systems by maximizing codebook utilization (Zhu et al., 2024; Dhariwal et al., 2020; Williams et al., 2020; Zheng & Vedaldi, 2023; Zhang et al., 2023; Ramesh et al., 2021). In this work, we contend that minimizing information loss is a more fundamental optimization objective. An intuitive rationale is that, for any information compression system, minimal information loss inherently promotes system stability (Touchette & Lloyd, 1999; Tomar et al., 2017; Touchette & Lloyd, 2001). We further formalize this relationship in Proposition 1, proving theoretically that minimizing information loss necessarily entails full codebook utilization, whereas the converse does not hold.

Let $X \sim P_X$ is defined on a measurable space $(\mathcal{X}, \mathcal{F})$. A deterministic quantizer with codebook size $K \in \mathbb{N}_+$ is a mapping $f : \mathcal{X} \to \{1, ..., K\}$ inducing $Z = f(X)$. We define the information loss $\mathcal{L}(f)$ and codebook utilization $\mathrm{U}(f)$ as:

$$\mathcal{L}(f) := \mathcal{H}(X \mid Z); \qquad \mathrm{U}(f) := \frac{\left|\{k \in \{1, ..., K\} : \ \mathbb{P}(Z = k) > 0\}\right|}{K}.$$

We first introduce a standard assumption in quantization analysis, which holds for continuous distributions with densities.

**Assumption 1** (Non-atomicity). *For any measurable set $A \subseteq \mathcal{X}$ with $\mathbb{P}(X \in A) > 0$, there exist disjoint measurable subsets $A_1, A_2 \subseteq A$ satisfying $\mathbb{P}(X \in A_i) > 0$ for $i = 1, 2$.*

**Proposition 1.** *(a) Let*

$$f^\star = \underset{f:\mathcal{X}\to\{1,...,K\}}{\arg\min} \ \mathcal{H}(X \mid f(X)).$$

*Under assumption 1, we have $\mathrm{U}(f^\star) = 1$.*
*(b) There exist quantizers $g : \mathcal{X} \to \{1, ..., K\}$ satisfying $\mathrm{U}(g) = 1$ that are not minimizers of the conditional entropy:*

$$\mathcal{H}(X \mid g(X)) > \underset{f:\mathcal{X}\to\{1,...,K\}}{\min} \ \mathcal{H}(X \mid f(X)).$$

The proposition implies that minimal information loss necessarily leads to 100% codebook utilization, while the converse does not hold. We provide the proof in Appendix A. Further in Appendix A, we specialize the notion of information loss to the Mean Squared Error (MSE) setting, showing in Proposition 4 that minimizing MSE guarantees 100% codebook utilization, while Proposition 5 demonstrates the existence of quantizers that achieve full codebook utilization fails to minimize MSE.

## 4.3 FAIR COMPARISON CONDITIONS

Direct comparison of quantization error is insufficient for evaluating the intrinsic effectiveness of quantization algorithms and may even yield paradoxical conclusions. This limitation arises from the linear scaling relationship between quantization error and latent feature distribution variance under optimal codebook conditions, as demonstrated in Figure 2 (complete data sources in Appendix D). Consequently, when latent feature variance remains uncontrolled, the error metric becomes dominated by variance-driven artifacts that mask true algorithmic performance. Therefore,

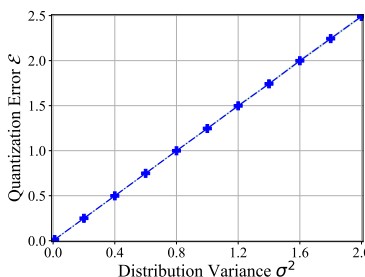

Figure 2: Linear relationship between quantization error $\mathcal{E}$ and distribution variance $\sigma^2$.

comparative evaluations of quantization algorithms must satisfy the normalization requirement specified in Condition 1 to accurately assess intrinsic algorithmic effectiveness.

**Condition 1.** *Latent feature distributions must be the same across all compared algorithms.*

Recent studies demonstrate that codebook size and token count significantly impact reconstruction performance (Yu et al., 2024; Zhu et al., 2024). As these two factors critically determine the compression ratio, they must be held constant across all quantization algorithms under comparison. Only under such strictly controlled conditions (as specified in Condition 2) can fair comparative evaluations be conducted to isolate the intrinsic effectiveness of different algorithms.

**Condition 2.** *Compression ratios must be held constant across all algorithms by using identical token counts and codebook sizes.*

### 4.4 QUANTIZATION ERROR ANALYSIS OF OPTIMAL VQ, PQ, AND SQ

In this section, we study the optimal quantization errors of idealized VQ, PQ, and SQ, and compare their performance under the same information constraints.

Given a probability distribution $\mathbb{P}$ over $\mathbb{R}^d$ and a codebook size $K \in \mathbb{N}_+$, we define the optimal quantization errors for VQ, PQ, and SQ as follows:

$$\mathcal{E}^*_{\mathrm{VQ}}(\mathbb{P}, K) := \inf_{\phi} \left\{ \mathbb{E}\big[\|X - \mathcal{Q}_\phi(X)\|_2^2\big] : |\phi| \leq K \right\},$$

$$\mathcal{E}^*_{\mathrm{PQ}}(\mathbb{P}, K, M) := \inf_{\phi} \left\{ \mathbb{E}\big[\|X - \mathcal{Q}_\phi(X)\|_2^2\big] : \phi = \bigoplus_{m=1}^{M} \phi_m, \ \phi_m \subseteq \mathbb{R}^{d_m}, |\phi_m| = n_m, \prod_{m=1}^{M} n_m \leq K \right\},$$

$$\mathcal{E}^*_{\mathrm{SQ}}(\mathbb{P}, K) := \inf_{\phi} \left\{ \mathbb{E}\big[\|X - \mathcal{Q}_\phi(X)\|_2^2\big] : \phi = \bigoplus_{m=1}^{d} \phi_m, \ \phi_m \subseteq \mathbb{R}, |\phi_m| = n_m, \prod_{m=1}^{d} n_m \leq K \right\}.$$

Here $|\phi|$ denotes the size of the codebook $\phi$, and $\bigoplus$ denotes the Cartesian product of sets.

Clearly, SQ is a special case of PQ with $M = d$, and PQ is a special case of VQ. Therefore, we have the following relationship among the optimal quantization errors:

$$\mathcal{E}^*_{\mathrm{VQ}}(\mathbb{P}, K) \leq \mathcal{E}^*_{\mathrm{PQ}}(\mathbb{P}, K, M) \leq \mathcal{E}^*_{\mathrm{SQ}}(\mathbb{P}, K), \quad \text{for any } M \in [d].$$

Let $\mathcal{P}$ be the set of all probability distributions over $[-1, 1]^d$. The following results provide quantitative characterizations of the optimal quantization errors for VQ, PQ, and SQ.

**Proposition 2.** *For any $K \geq 2^d$, we have*

$$\frac{d}{4K^{2/d}} \leq \sup_{\mathbb{P} \in \mathcal{P}} \mathcal{E}^*_{VQ}(\mathbb{P}, K) \leq \sup_{\mathbb{P} \in \mathcal{P}} \mathcal{E}^*_{SQ}(\mathbb{P}, K) \leq \frac{8d}{K^{2/d}}$$

See Appendix B for the proof. Proposition 2 shows that for worst-case distributions, the optimal quantization errors of VQ, PQ, and SQ are the same up to universal constant factors. All three methods achieve a quantization error that scales as $\Theta(d/K^{2/d})$.

On the other hand, when the data distribution has intrinsic low-dimensional structures, VQ can significantly outperform PQ and SQ. We illustrate this phenomenon in the following proposition.

**Proposition 3.** *If the support of $\mathbb{P} \in \mathcal{P}$ is contained in a $d_{eff}$-dimensional subspace of $\mathbb{R}^d$ with $d_{eff} < d$, then for any $K \geq 2^{d_{eff}}$, we have*

$$\mathcal{E}^*_{VQ}(\mathbb{P}, K) \leq \frac{8dd_{eff}}{K^{2/d_{eff}}}.$$

*On the other hand, there exists a 1-dimensional linear subspace $L \subseteq \mathbb{R}^d$ and a distribution $\mathbb{P}$ supported on $L \cap [-1, 1]^d$ such that for any $K \geq 2^d$, we have*

$$\mathcal{E}^*_{PQ}(\mathbb{P}, K, M) \geq \frac{M}{4} K^{-2/M}, \quad \text{for any } M \in [d], \quad \text{and} \quad \mathcal{E}^*_{SQ}(\mathbb{P}, K) \geq \frac{d}{4} K^{-2/d}.$$

See Appendix C for the proof. Proposition 3 shows that when the data distribution has an intrinsic dimension $d_{\mathrm{eff}} < d$, VQ can achieve a quantization error that scales as $\mathcal{O}(K^{-2/d_{\mathrm{eff}}})$, which can be significantly smaller than the worst-case rate of $\Theta(d/K^{2/d})$ when $d_{\mathrm{eff}} \ll d$. On the other hand, with a simple 1-dimensional data distribution, SQ still suffers from the worst-case rate of $\Omega(d/K^{2/d})$ while VQ can achieve an $\mathcal{O}(dK^{-2})$ rate. The error of PQ interpolates between these two extremes, and its performance depends on the number of blocks $M$. Moreover, though the results are stated for linear subspaces, they can be easily extended to nonlinear manifolds using covering number arguments.

## 5 EMPIRICAL STUDY

To empirically validate our theoretical conclusions, we analyze three distinct quantization algorithms (VQ, PQ, and SQ) on ImageNet-1K (Deng et al., 2009) using the VQ-Transplant framework (Anonymous, 2025). This framework offers two critical advantages: (1) systematic elimination of confounding factors through strict controls (Sec. 3), and (2) enforcement of identical latent feature distributions across all algorithms during optimization, thereby satisfying Condition 1 (Sec. 4.3). By additionally holding codebook size and token count constant, this design ensures a fair comparison.

### 5.1 EXPERIMENTAL SETUP

**VQ-Transplant Framework.** We employ a pre-trained VAR tokenizer (Tian et al., 2024) for all experiments to implement VQ-Transplant. The VQ-Transplant framework operates through two distinct stages: VQ Module Substitution and Decoder Adaptation. During the VQ module substitution stage, we freeze the parameters of the pre-trained encoder-decoder and replace its native VQ module with newly introduced quantization modules. Subsequently, during decoder adaptation, we freeze the parameters of both the encoder and transplanted VQ modules, while updating the decoder parameters to align feature priors with the new quantization space. Further implementation details of the VQ-Transplant framework are provided in Appendix E.

**Quantization Algorithms.** We examine the intrinsic effectiveness of three quantization paradigms: VQ, PQ, and SQ. For VQ, we evaluate five variants: Vanilla VQ (van den Oord et al., 2017), EMA VQ (Razavi et al., 2019), Online VQ (Zheng & Vedaldi, 2023), Wasserstein VQ (Fang et al., 2025), and MMD VQ (Anonymous, 2025) (see Appendix F for methodological details). For PQ, we implement five corresponding variants: Vanilla VP2, EMA VP2, Online VP2, Wasserstein VP2, and MMD VP2. For SQ, we employ three methods: FSQ (Mentzer et al., 2024a), LFQ (Yu et al., 2024), and BSQ (Zhao et al., 2025). All methods use identical token counts and codebook sizes, satisfying Condition 2. Implementation details and training protocols are documented in Appendix G.

**Evaluation Metrics.** Following VQ-Transplant (Anonymous, 2025), we report quantization error ($\mathcal{E}$) and codebook utilization rate (U) to evaluate quantization performance. To assess reconstruction quality, we report Peak Signal-to-Noise Ratio (PSNR), Structural Similarity Index (SSIM), Fréchet Inception Distance (r-FID) (Heusel et al., 2017), Learned Perceptual Image Patch Similarity (LPIPS) (Zhang et al., 2018), and Inception Score (r-IS) (Salimans et al., 2016).

### 5.2 EXPERIMENTAL RESULTS

**Comparison among VQ, PQ, and SQ.** As evidenced in Table 3, when substituting VQ modules, the optimal VQ approach (MMD VQ) achieves the lowest quantization error, succeeded by the optimal PQ method (EMA VP2), with the optimal SQ technique (BSQ) yielding the highest error. This demonstrates VQ's superior representational capacity regarding optimal quantization performance, empirically validating our theoretical analysis in Section 4.4.

As established in Section 4.4, since VQ generalizes PQ and PQ generalizes SQ, VQ methods naturally possess greater modeling flexibility and higher performance potential. Crucially, during decoder adaptation, MMD VQ's enhanced information preservation translates to state-of-the-art reconstruction quality as measured by r-FID. Furthermore, VQ methods consistently outperform alternatives across most reconstruction metrics, with codebook utilization presenting the sole exception where PQ methods exhibit superior performance.

**Correlation Analyses.** To empirically demonstrate that minimizing quantization error is a more critical optimization objective than maximizing codebook utilization, we compute Spearman's rank correlations between each objective and reconstruction fidelity (r-FID) using the data from Table 3. Our analysis reveals a near-perfect, statistically significant positive correlation between quantization error $\mathcal{E}$ and r-FID ($\rho = 0.996$, $p < 10^{-5}$), indicating that higher $\mathcal{E}$ is strongly associated with degraded reconstruction quality. In contrast, codebook utilization U exhibits only a moderate negative correlation with r-FID ($\rho = -0.650$, $p = 0.016$), suggesting a weaker relationship where increased U is modestly associated with improved performance. These results demonstrate that $\mathcal{E}$ substantially outweighs U in importance for reconstruction quality. Notably, the strength of the ($\mathcal{E}$, r-FID) correlation ($\rho = 0.996$, $p < 10^{-5}$) establishes minimizing quantization error as the paramount

Table 3: Comparative reconstruction performance of VQ, PQ, and SQ quantization methods on ImageNet-1K. †: Results cited from VQ-Transplant (Anonymous, 2025). Within each quantization type and phase (Substitution/Adaptation), optimal values are underlined; overall best results per metric are **bold**.

| Approaches | Types | Phase | Tokens | $K$ | $\mathcal{E}(\downarrow)$ | U ($\uparrow$) | PSNR($\uparrow$) | SSIM($\uparrow$) | LPIPS ($\downarrow$) | r-FID($\downarrow$) | r-IS($\uparrow$) |
|---|---|---|---|---|---|---|---|---|---|---|---|
| Vanilla VQ† | VQ | Substitution | 512 | 65536 | 0.422 | 0.2% | 22.04 | 53.1 | 0.243 | 10.89 | 103.8 |
| EMA VQ† | VQ | Substitution | 512 | 65536 | 0.217 | 65.5% | 24.94 | 65.9 | 0.127 | 1.78 | 185.8 |
| Online VQ† | VQ | Substitution | 512 | 65536 | 0.280 | 13.5% | 24.42 | 63.2 | 0.147 | 2.28 | 174.2 |
| Wasserstein VQ† | VQ | Substitution | 512 | 65536 | **0.201** | 99.6% | 25.22 | **66.9** | **0.121** | 1.76 | 186.0 |
| MMD VQ† | VQ | Substitution | 512 | 65536 | **0.201** | 99.9% | **25.24** | 66.8 | **0.121** | 1.69 | 187.3 |
| Vanilla VP2 | PQ | Substitution | 512 | 65536 | 0.233 | 59.2% | 24.80 | 65.3 | 0.130 | 1.84 | 183.1 |
| EMA VP2 | PQ | Substitution | 512 | 65536 | 0.209 | **100%** | 25.08 | 66.4 | 0.123 | 1.68 | 187.2 |
| Online VP2 | PQ | Substitution | 512 | 65536 | 0.211 | **100%** | 25.09 | 66.3 | 0.124 | 1.79 | 185.7 |
| Wasserstein VP2 | PQ | Substitution | 512 | 65536 | 0.217 | **100%** | 24.79 | 65.8 | 0.128 | 1.78 | 185.7 |
| MMD VP2 | PQ | Substitution | 512 | 65536 | 0.212 | **100%** | 25.00 | 66.1 | 0.123 | **1.61** | **189.5** |
| FSQ | SQ | Substitution | 512 | 65536 | 0.300 | 71.0% | 23.85 | 60.6 | 0.157 | 2.79 | 167.2 |
| LFQ | SQ | Substitution | 512 | 65536 | 0.279 | 29.8% | 24.06 | 62.6 | 0.146 | 2.15 | 176.2 |
| BSQ | SQ | Substitution | 512 | 65536 | 0.231 | **100%** | 24.55 | 65.2 | 0.132 | 1.96 | 182.1 |
| Vanilla VQ† | VQ | Adaptation | 512 | 65536 | 0.422 | 0.2% | 21.19 | 50.7 | 0.209 | 5.05 | 118.9 |
| EMA VQ† | VQ | Adaptation | 512 | 65536 | 0.217 | 65.5% | 24.36 | 64.1 | 0.111 | 0.99 | 194.3 |
| Online VQ† | VQ | Adaptation | 512 | 65536 | 0.280 | 13.5% | 23.84 | 61.6 | 0.130 | 1.38 | 182.9 |
| Wasserstein VQ† | VQ | Adaptation | 512 | 65536 | **0.201** | 99.6% | **24.68** | **65.4** | **0.106** | 0.92 | 195.5 |
| MMD VQ† | VQ | Adaptation | 512 | 65536 | **0.201** | 99.9% | 24.65 | 65.0 | **0.106** | **0.86** | **197.1** |
| Vanilla VP2 | PQ | Adaptation | 512 | 65536 | 0.233 | 59.2% | 24.28 | 64.0 | 0.114 | 1.07 | 191.7 |
| EMA VP2 | PQ | Adaptation | 512 | 65536 | 0.209 | **100%** | 24.55 | 64.9 | 0.107 | 0.93 | 195.4 |
| Online VP2 | PQ | Adaptation | 512 | 65536 | 0.211 | **100%** | 24.53 | 64.7 | 0.108 | 0.95 | 195.3 |
| Wasserstein VP2 | PQ | Adaptation | 512 | 65536 | 0.217 | **100%** | 24.44 | 64.6 | 0.110 | 0.99 | 193.5 |
| MMD VP2 | PQ | Adaptation | 512 | 65536 | 0.212 | **100%** | 24.43 | 64.5 | 0.109 | 0.95 | 196.1 |
| FSQ | SQ | Adaptation | 512 | 65536 | 0.300 | 71.0% | 23.27 | 59.1 | 0.134 | 1.52 | 179.3 |
| LFQ | SQ | Adaptation | 512 | 65536 | 0.279 | 29.8% | 23.42 | 60.7 | 0.130 | 1.30 | 183.2 |
| BSQ | SQ | Adaptation | 512 | 65536 | 0.231 | **100%** | 24.06 | 63.6 | 0.117 | 1.07 | 190.8 |

objective for achieving high reconstruction fidelity, superseding the role of codebook utilization. This empirical evidence strongly supports our theoretical framework presented in Section 4.2.

**Analyses on Codebook Size and Token Count.** As demonstrated in Table 4 in Appendix H, we scale the codebook size incrementally by a factor of 2, from 1024 to 65536. Substituting the VQ modules resulted in a reduction of the quantization error $\mathcal{E}$ from 0.318 to 0.201. Additionally, after decoder adaptation, the critical reconstruction metric, r-FID, improved from 1.90 to 0.86. These findings indicate that the codebook size has a moderate yet significant impact on both $\mathcal{E}$ and r-FID. In contrast, the token count exhibits a more pronounced effect on these metrics. As illustrated in Table 5 (Appendix H), doubling the token count from 256 to 1024 led to a substantial decrease in quantization error from 0.369 to 0.035, while r-FID improved significantly from 3.06 to 0.42.

**Equivalence Between Token Count and Codebook Size.** To further investigate the relationship between token count and codebook size, we compared two scenarios: doubling the token count versus squaring the codebook size, as shown in Table 6 in Appendix H. These two scenarios exhibited nearly identical performance, which provides empirical support for the equivalence relationship predicted in Section 4.1, specifically in Finding 1: doubling the token count is equivalent to squaring the codebook size in terms of information capacity.

## 6 CONCLUSION

In this paper, we proposed a unifying conceptual framework that elucidates the impact and tradeoffs of different quantization objectives. By viewing quantization as information compression, we resolve longstanding ambiguities regarding quantization objectives and comparative algorithmic effectiveness. Our empirical and theoretical analysis conclusively establishes that minimizing quantization error, rather than maximizing codebook utilization, is the paramount optimization objective for ensuring reconstruction fidelity and training stability. To enable artifact-free comparisons, we introduced two critical fairness conditions: identical latent feature distributions and compression ratios. Under these conditions, our empirical evaluation demonstrates the superiority of modern VQ algorithms over SQ/PQ baselines in minimizing information loss. Collectively, this work bridges persistent conceptual gaps in quantization theory and establishes the first principled methodology for artifact-free algorithmic evaluation. Our findings provide a robust and principled approach to understanding and optimizing quantization algorithms and paves the way for future advancements in the field.

## 7 REPRODUCIBILITY STATEMENT

To ensure full reproducibility of our results, the following resources are included in the supplemental materials: (1) complete training and evaluation source code, (2) execution scripts for all experiments, (3) comprehensive training logs capturing model dynamics, and (4) final model outputs and evaluation artifacts. To further support the research community, all resources—including pre-trained model weights, detailed documentation, and configuration files—will be publicly released on GitHub. This release will enable independent verification of our findings and facilitate future research.

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

# APPENDIX

## A PROOF OF PROPOSITION 1 AND EXTENSION TO MEAN SQUARED ERROR AS DISTORTION METRIC

*Proof.* Part (a): We prove by contradiction. Suppose $f^\star$ uses only $M < K$ codewords, leaving at least one codeword unused. Then there exists $k$ with $\mathbb{P}(Z = k) > 0$. Define

$$A^\star := \{x \in \mathcal{X} : f^\star(x) = k\}.$$

By Assumption 1, partition $A^\star$ into disjoint measurable sets $A_1, A_2$ with $\mathbb{P}(X \in A_i) > 0$. Construct a modified quantizer:

$$f'(x) = \begin{cases} f^\star(x), & x \notin A^\star \\ k, & x \in A_1 \\ k_{\text{new}}, & x \in A_2, \end{cases}$$

where $k_{\text{new}}$ is an unused codeword. Since $\sigma(f^\star(X)) \subsetneq \sigma(f'(X))$ and the conditional distributions on $A_1$ and $A_2$ differ, we have the strict entropy reduction:

$$\mathcal{H}(X \mid f'(X)) < \mathcal{H}(X \mid f^\star(X)).$$

This contradicts the optimality of $f^\star$. Thus any minimizer must satisfy $\mathrm{U}(f^\star) = 1$.

Part (b): Consider a discrete probability space $(\mathcal{X}, \mathcal{F}, \mathbb{P})$ with $\mathcal{X} = \{1, 2, 3, 4\}$, $\mathcal{F} = 2^{\mathcal{X}}$, and probability measure:

$$\mathbb{P}(X = 1) = \frac{1}{2}, \quad \mathbb{P}(X = 2) = \frac{1}{4}, \quad \mathbb{P}(X = 3) = \mathbb{P}(X = 4) = \frac{1}{8}.$$

Set $K = 2$. Let $f^\star$ be the $\sigma$-measurable quantizer partitioning the state space as:

$$f^\star(x) = \begin{cases} 1 & x = 1 \\ 2 & x \in \{2, 3, 4\} \end{cases} \quad \text{with cells} \quad \mathcal{C}_1^\star = \{1\}, \ \mathcal{C}_2^\star = \{2, 3, 4\}.$$

The conditional entropy is:

$$\mathcal{H}(X \mid f^\star(X)) = \sum_{k=1}^2 \mathbb{P}(\mathcal{C}_k^\star)\mathcal{H}(X \mid \mathcal{C}_k^\star) = \frac{1}{2} \cdot 0 + \frac{1}{2}\left(-\sum_{x=2}^4 \frac{\mathbb{P}(x)}{\mathbb{P}(\mathcal{C}_2^\star)} \log_2 \frac{\mathbb{P}(x)}{\mathbb{P}(\mathcal{C}_2^\star)}\right) = \frac{3}{4} \text{ bits.}$$

This achieves the minimum by exhaustive enumeration of partitions.

Now define $g(x) = \mathbf{1}_{\{1,2\}}(x) + 2 \cdot \mathbf{1}_{\{3,4\}}(x)$ with cells:

$$\mathcal{C}_1^g = \{1, 2\}, \quad \mathcal{C}_2^g = \{3, 4\}.$$

This satisfies $\mathrm{U}(g) = 1$ since $\mathbb{P}(\mathcal{C}_1^g) = \frac{3}{4} > 0$ and $\mathbb{P}(\mathcal{C}_2^g) = \frac{1}{4} > 0$. However:

$$\mathcal{H}(X \mid g(X)) = \frac{3}{4}\mathcal{H}(X \mid \mathcal{C}_1^g) + \frac{1}{4}\mathcal{H}(X \mid \mathcal{C}_2^g)$$

where

$$\mathcal{H}(X \mid \mathcal{C}_1^g) = -\left(\frac{2}{3}\log_2\frac{2}{3} + \frac{1}{3}\log_2\frac{1}{3}\right) = \log_2 3 - \frac{2}{3}, \quad \mathcal{H}(X \mid \mathcal{C}_2^g) = 1.$$

Thus:

$$\mathcal{H}(X \mid g(X)) = \frac{3}{4}\left(\log_2 3 - \frac{2}{3}\right) + \frac{1}{4} = \frac{3}{4}\log_2 3 - \frac{1}{2} \approx 0.939 > 0.75 = \mathcal{H}(X \mid f^\star(X)).$$

This completes the proof. Hence, full utilization alone does not guarantee minimal information loss, as demonstrated by the quantizer $g$. $\qquad\square$

**Remark 1.** *If randomized mapping are allowed, one may enforce $\mathbb{P}(Z = k) = 1/K > 0$ nearly independent of $X$, so $\mathrm{U} = 1$ while $\mathcal{I}(X; Z) \approx 0$ and $\mathcal{H}(X \mid Z) \approx \mathcal{H}(X)$ is maximal. This further shows that full utilization is far from sufficient.*

We then specialize the notion of information loss to the Mean Squared Error (MSE) setting, showing in Proposition 4 that minimizing MSE guarantees 100% codebook utilization, while Proposition 5 demonstrates the existence of quantizers that achieve full codebook utilization fails to minimize MSE.

**Mean Squared Error as Distortion Metric.** Define a reconstruction mapping $\widehat{x} : \{1, \ldots, K\} \to \mathbb{R}^d$. The mean squared error (MSE) distortion for a quantizer $f$ paired with this reconstruction mapping is given by:

$$\mathcal{E}(f, \widehat{x}) := \mathbb{E}\left[\, \|X - \widehat{x}(f(X))\|^2 \,\right].$$

**Proposition 4.** *Let*

$$f^\star, \widehat{x}^\star = \operatorname*{arg\,min}_{f:\mathcal{X}\to\{1,\ldots,K\},\widehat{x}} \mathcal{E}(f, \widehat{x}),$$

*then we have* $\mathrm{U}(f^\star) = 1$.

*Proof.* We prove by contradiction. Suppose that $(f^\star, \widehat{x}^\star)$ is optimal but $\mathrm{U}(f^\star) < 1$. Then there exists a codeword $k \in \{1, \ldots, K\}$ such that $\mathbb{P}(f^\star(X) = k) = 0$.

By the optimality of $\widehat{x}^\star$, we have for each $k$ that

$$\widehat{x}^\star(k) = \mathbb{E}[X \mid f^\star(X) = k].$$

Since $\mathbb{P}(f^\star(X) = k) = 0$, the value of $\widehat{x}^\star(k)$ is arbitrary.

Now, by Assumption 1, there exists a cell $A^\star := \{x \in \mathcal{X} : f^\star(x) = k^\star\}$ with $\mathbb{P}(X \in A^\star) > 0$ that can be partitioned into two disjoint subsets $A_1$ and $A_2$ such that $\mathbb{P}(X \in A_1) > 0$ and $\mathbb{P}(X \in A_2) > 0$.

Define a refined $f'$ by

$$f'(x) = \begin{cases} k^\star & \text{if } x \in A_1, \\ k & \text{if } x \in A_2, \\ f^\star(x) & \text{otherwise.} \end{cases}$$

Define the $\widehat{x}'$ optimally as

$$\widehat{x}'(z) = \mathbb{E}[X \mid f'(X) = z].$$

By the law of total variance, we have

$$\mathcal{E}(f', \widehat{x}') = \mathbb{E}\left[\mathrm{Var}(X \mid f'(X))\right].$$

Since the partition of $A^\star$ into $A_1$ and $A_2$ is nontrivial, we obtain

$$\mathbb{E}\left[\mathrm{Var}(X \mid f'(X))\right] < \mathbb{E}\left[\mathrm{Var}(X \mid f^\star(X))\right].$$

This implies $\mathcal{E}(f', \widehat{x}') < \mathcal{E}(f^\star, \widehat{x}^\star)$, contradicting the optimality of $(f^\star, \widehat{x}^\star)$. Therefore, $\mathrm{U}(f^\star) = 1$. $\qquad\square$

**Proposition 5.** *There exist deterministic quantizers $f$ with $\mathrm{U}(f) = 1$ such that, even with the optimal reconstruction mapping*

$$\widehat{x}_f(z) = \mathbb{E}[X \mid f(X) = z],$$

*the MSE distortion*

$$\mathcal{E}(f, \widehat{x}_f) = \mathbb{E}\left[\|X - \widehat{x}_f(f(X))\|^2\right]$$

*is strictly larger than the global minimum* $\min_{g,\widehat{x}} \mathcal{E}(g, \widehat{x})$.

*Proof.* Consider a random variable $X$ supported on two well-separated clusters in $\mathbb{R}^d$ with positive probability masses, denoted as $\mathcal{C}_1$ and $\mathcal{C}_2$. Let the codebook size be $K = 2$. Define the optimal quantizer $f^\star$ as

$$f^\star(x) = \begin{cases} 1 & \text{if } x \in \mathcal{C}_1, \\ 2 & \text{if } x \in \mathcal{C}_2, \end{cases}$$

with the corresponding optimal reconstruction mapping

$$\widehat{x}^\star(z) = \mathbb{E}[X \mid f^\star(X) = z].$$

Since $f^\star$ assigns each cluster to a distinct codeword, the reconstruction points coincide with the cluster means, and the resulting MSE distortion

$$\mathcal{E}(f^\star, \widehat{x}^\star) = \mathbb{E}[\mathrm{Var}(X \mid f^\star(X))]$$

is small.

Now construct another quantizer $f$ that satisfies $U(f) = 1$ but mixes the clusters. Specifically, partition each cluster $\mathcal{C}_i$ ($i = 1, 2$) into two subsets $\mathcal{C}_i^1$ and $\mathcal{C}_i^2$ with equal probability mass, and define

$$f(x) = \begin{cases} 1 & \text{if } x \in \mathcal{C}_1^1 \cup \mathcal{C}_2^1, \\ 2 & \text{if } x \in \mathcal{C}_1^2 \cup \mathcal{C}_2^2. \end{cases}$$

The optimal reconstruction mapping for $f$ is

$$\widehat{x}_f(z) = \mathbb{E}[X \mid f(X) = z].$$

However, since each codeword now contains points from both clusters, the reconstruction points are placed near the global means of the mixed subsets rather than the cluster means. This leads to increased conditional variance within each cell. By the law of total variance, we have

$$\mathcal{E}(f, \widehat{x}_f) = \mathbb{E}[\text{Var}(X \mid f(X))] > \mathbb{E}[\text{Var}(X \mid f^\star(X))] = \mathcal{E}(f^\star, \widehat{x}^\star).$$

Thus, full utilization ($U(f) = 1$) does not guarantee minimal MSE distortion. $\qquad\square$

**Remark 2.** *The construction extends to any $K \geq 2$ by mixing portions of at least two well-separated regions across multiple codewords. The suboptimality is strict whenever the merged parts have different conditional means.*

## B    PROOF OF PROPOSITION 2

We first prove the upper bound by constructing an SQ scheme. Let $n = \lfloor K^{1/d} \rfloor$. We construct the codebook $\phi = \bigoplus_{m=1}^d \phi_m$ with $\phi_m = \left\{ -1 + \frac{2i}{n-1} \; : \; i = 0, 1, \ldots, n-1 \right\}$ for each $m \in [d]$. Clearly, we have $|\phi| = n^d \leq K$. For any $x = (x_1, x_2, \ldots, x_d) \in [-1, 1]^d$, let $\mathcal{Q}_\phi(x) = (\mathcal{Q}_{\phi_1}(x_1), \mathcal{Q}_{\phi_2}(x_2), \ldots, \mathcal{Q}_{\phi_d}(x_d))$, where $\mathcal{Q}_{\phi_m}(x_m)$ is the closest point in $\phi_m$ to $x_m$. Then we have

$$\mathbb{E}\big[\|X - \mathcal{Q}_\phi(X)\|_2^2\big] \leq \sup_{x \in [-1,1]^d} \|x - \mathcal{Q}_\phi(x)\|_2^2 \leq d \cdot \left(\frac{2}{n-1}\right)^2 \leq \frac{4d}{(K^{1/d} - 1)^2} \leq \frac{8d}{K^{2/d}}.$$

On the other hand, we prove the lower bound by estimating the optimal VQ error for the uniform distribution $\mathbb{P} = \text{Unif}([-1, 1]^d) \in \mathcal{P}$. Given a codebook $\phi = \{e_1, e_2, \ldots, e_K\}$, we define the set

$$\bar{S}_\phi(r) := [-1, 1]^d \setminus \bigcup_{k=1}^K B(e_k, r).$$

It is easy to see that

$$\mathbb{P}\big(X \in \bar{S}_\phi(r)\big) = \frac{|\bar{S}_\phi(r)|}{2^d} \geq 1 - K \cdot \frac{\pi^{d/2} r^d}{2^d \Gamma(d/2 + 1)} \geq 1 - K\left(\frac{r}{\sqrt{d}}\right)^d.$$

Choosing $r = \sqrt{d} \cdot (2K)^{-1/d}$, we have $\mathbb{P}\big(X \in \bar{S}_\phi(r)\big) \geq 1/2$. For any $x \in \bar{S}_\phi(r)$, we have $\|x - \mathcal{Q}_\phi(x)\|_2 > r$. Therefore, we obtain

$$\mathbb{E}\big[\|X - \mathcal{Q}_\phi(X)\|_2^2\big] \geq \mathbb{E}\big[\|X - \mathcal{Q}_\phi(X)\|_2^2 \mid X \in \bar{S}_\phi(r)\big] \cdot \mathbb{P}\big(X \in \bar{S}_\phi(r)\big)$$

$$\geq r^2 \cdot \mathbb{P}\big(X \in \bar{S}_\phi(r)\big) \geq \frac{d}{4K^{2/d}}.$$

## C    PROOF OF PROPOSITION 3

We first prove the upper bound for VQ. By assumption, there exists a $d_{\text{eff}}$-dimensional subspace $L \subseteq \mathbb{R}^d$ such that $\mathbb{P}$ is supported on $L \cap [-1, 1]^d$. Let $U \in \mathbb{R}^{d \times d_{\text{eff}}}$ be an orthonormal basis of $L$. Then for any $x \in L$, we can write $x = Uz$ for some $z \in \mathbb{R}^{d_{\text{eff}}}$. Let $\tilde{\mathbb{P}}$ be the distribution of $Z$ when $X \sim \mathbb{P}$. We note that

$$\sup_{z \in \text{supp}(\tilde{\mathbb{P}})} \|z\|_\infty \leq \sup_{z \in \text{supp}(\tilde{\mathbb{P}})} \|z\|_2 = \sup_{x \in \text{supp}(\mathbb{P})} \|U^\top x\|_2 \leq \sup_{x \in \text{supp}(\mathbb{P})} \|x\|_2 \leq \sqrt{d}.$$

Therefore, we can view $\tilde{\mathbb{P}}$ as a distribution over $[-\sqrt{d}, \sqrt{d}]^{d_{\text{eff}}}$. Let $n = \lfloor K^{1/d_{\text{eff}}} \rfloor$. Invoking the same SQ scheme as in the proof of Proposition 2 for the distribution $\tilde{\mathbb{P}}$, we can construct a codebook $\tilde{\phi} = \bigoplus_{m=1}^{d_{\text{eff}}} \tilde{\phi}_m$ with $\left|\tilde{\phi}\right| \leq K$ such that

$$\mathbb{E}_{Z \sim \tilde{\mathbb{P}}}\big[\|Z - \mathcal{Q}_{\tilde{\phi}}(Z)\|_2^2\big] \leq \frac{8 d d_{\text{eff}}}{K^{2/d_{\text{eff}}}}.$$

Turning to the error lower bound for PQ and SQ, we consider the 1-dimensional subspace $L = \{\alpha \mathbf{1} \,:\, \alpha \in \mathbb{R}\} \subseteq \mathbb{R}^d$, where $\mathbf{1} = (1, 1, \ldots, 1)^\top \in \mathbb{R}^d$. Let $\mathbb{P}$ be the uniform distribution over $L \cap [-1, 1]^d$. Given any PQ codebook $\phi = \bigoplus_{m=1}^M \phi_m$ with $|\phi| \leq K$, for each $m \in [M]$, the $m$-th sub-codebook $\phi_m$ contains $n_m$ points in $\mathbb{R}^{d_m}$. Let $\Pi_m : \mathbb{R}^d \to \mathbb{R}^{d_m}$ be the projection operator that extracts the coordinates in the $m$-th subspace. The sub-codebook $\phi_m$ solves the VQ problem for the probability distribution of $\Pi_m(X)$, where $X \sim \mathbb{P}$. Since $\mathbb{P}$ is uniform on $L \cap [-1, 1]^d$, the distribution of $\Pi_m(X)$ is uniform on $\Pi_m\big(L \cap [-1, 1]^d\big)$, which is a line segment with length $2\sqrt{d_m}$. Therefore, by the same argument as in the proof of Proposition 2, we have

$$\mathbb{E}\big[\|\Pi_m(X) - \mathcal{Q}_{\phi_m}(\Pi_m(X))\|_2^2\big] \geq \frac{d_m}{4 n_m^2}.$$

Aggregating the errors over all subspaces, we obtain

$$\mathbb{E}\big[\|X - \mathcal{Q}_\phi(X)\|_2^2\big] = \sum_{m=1}^M \mathbb{E}\big[\|\Pi_m(X) - \mathcal{Q}_{\phi_m}(\Pi_m(X))\|_2^2\big] \geq \sum_{m=1}^M \frac{d_m}{4 n_m^2} \geq \frac{M}{4} K^{-2/M}.$$

In particular, for SQ with $M = d$, we have

$$\mathbb{E}\big[\|X - \mathcal{Q}_\phi(X)\|_2^2\big] \geq \frac{d}{4} K^{-2/d},$$

which completes the proof.

# D  EXPERIMENTAL DETAILS IN SECTION 4.3

In this section, we analyze the relationship between quantization error and latent feature distribution variance under optimal codebook conditions. As demonstrated in Fang et al. (2025), minimal quantization error is achieved when features and codebook vectors are identically distributed. Therefore, we maintain identical distributions for feature and codebook vectors in our simulation analyses. Specifically, we sample feature vectors $\{\boldsymbol{z}_i\}_{i=1}^N$ and code vectors $\{\boldsymbol{e}_k\}_{k=1}^K$ from $\mathcal{N}_d(\boldsymbol{0}, \sigma^2 \boldsymbol{I})$, varying $\sigma^2 \in \{0.01, 0.2, 0.4, 0.6, 0.8, 1.0, 1.2, 1.4, 1.6, 1.8, 2.0\}$ with $K = 8192$, $d = 8$, and $N = 100000$. Each synthetic experiment undergoes five independent trials, with averaged results shown in Figure 2. We observe a pronounced linear relationship between quantization error and distribution variance.

# E  VQ-TRANSPLANT: EFFICIENT REPLACEMENT OF VECTOR QUANTIZATION MODULES

To mitigate the computational overhead of end-to-end retraining, Anonymous (2025) propose **VQ-Transplant**, a framework that efficiently replaces vector quantization (VQ) modules within pre-trained visual tokenizers. This approach operates in two distinct stages to maintain model performance while replacing fundamental components.

**Stage I: VQ Module Substitution.**  Given a pre-trained discrete visual tokenizer with encoder $\mathcal{E}_{\theta^*}$, decoder $\mathcal{D}_{\varphi^*}$, and original VQ module $\mathcal{Q}_{\phi^*}^{\text{pretrain}}$, VQ-Transplant substitutes $\mathcal{Q}_{\phi^*}^{\text{pretrain}}$ with a new VQ module $\mathcal{Q}_\phi^{\text{new}}$ while keeping $\theta^*$ and $\varphi^*$ frozen. For an input image $\boldsymbol{x}$, the encoder produces latent embeddings $\boldsymbol{z}_e = \mathcal{E}_{\theta^*}(\boldsymbol{x})$, which are then quantized by the new VQ module as $\boldsymbol{z}_q(\phi) = \mathcal{Q}_\phi^{\text{new}}(\boldsymbol{z}_e)$. The optimization objective for the new VQ module is:

$$\mathcal{L}_{\text{VQ}}(\phi) = \|\text{sg}(\boldsymbol{z}_e) - \boldsymbol{z}_q(\phi)\|_2^2 + \gamma \mathcal{L}_{\text{unique}}(\mathcal{Q}_\phi^{\text{new}}), \tag{10}$$

where $\mathcal{L}_{\text{unique}}$ enforces codebook uniqueness constraints (e.g., Wasserstein loss for Wasserstein VQ (Fang et al., 2025)) and $\gamma$ balances the loss terms. This stage minimizes quantization error while satisfying the new VQ algorithm's inherent constraints.

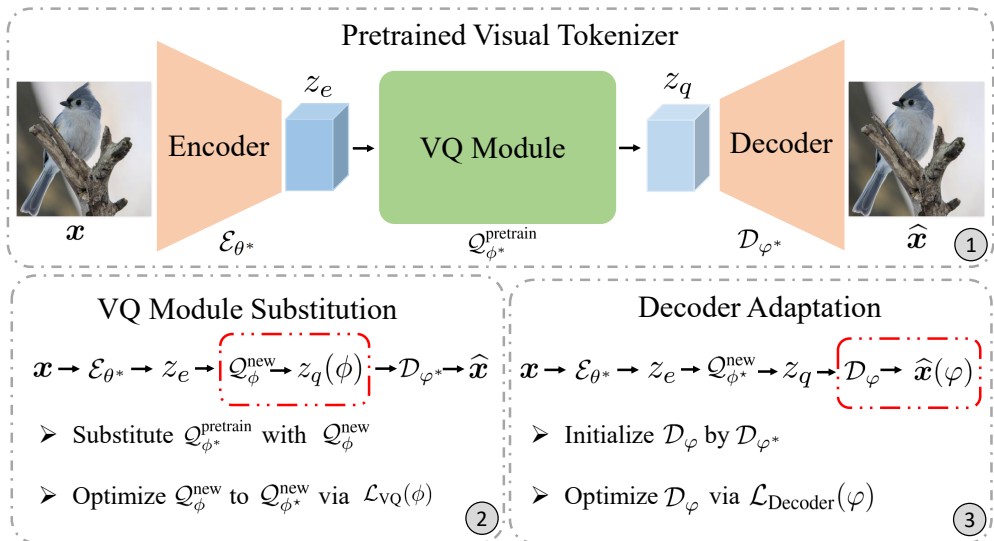

Figure 3: The overall illustration of the **VQ-Transplant**. Block 1 represents a pretrained visual tokenizer which comprises three key components: an encoder, decoder and native VQ module. Block 2 and 3 denote the VQ module substitution and decoder adaptation stages in the VQ-Transplant framework.

**Stage II: Decoder Adaptation.** Although Stage I reduces quantization error, the frozen decoder $\mathcal{D}_{\varphi^*}$ remains suboptimal for reconstructing inputs from $z_q(\phi)$ due to decoder-quantization space mismatch. To address this, VQ-Transplant employs a lightweight decoder adaptation scheme. With encoder $\mathcal{E}_{\theta^*}$ and optimized VQ module $\mathcal{Q}_{\phi^*}^{\text{new}}$ remaining frozen, the decoder parameters $\varphi$ (initialized from $\varphi^*$) are updated. The reconstruction pipeline becomes $\widehat{\boldsymbol{x}}(\varphi) = \mathcal{D}_{\varphi}(\mathcal{Q}_{\phi^*}^{\text{new}}(\mathcal{E}_{\theta^*}(\boldsymbol{x})))$. The decoder is optimized via:

$$\mathcal{L}_{\text{Decoder}}(\varphi) = \|\widehat{\boldsymbol{x}}(\varphi) - \boldsymbol{x}\|_2^2 + \lambda_P \mathcal{L}_{\text{Per}}(\varphi) + \lambda_G \mathcal{L}_{\text{GAN}}(\varphi), \quad (11)$$

where hyperparameters $\lambda_P$ and $\lambda_G$ balance the terms. Following Tian et al. (2024), the method adopts a frozen DINO-S (Caron et al., 2021; Oquab et al., 2024) discriminator with StyleGAN-like architecture (Karras et al., 2020; 2019). Discriminator training incorporates DiffAug (Zhao et al., 2020), consistency regularization (Zhang et al., 2019), and LeCAM regularization (Tseng et al., 2021) as implemented in Tian et al. (2024).

# F  MMD VQ: A New VQ algorithm compatible with the VQ-transplant framework

To improve compatibility with the VQ-Transplant framework, Anonymous (2025) introduce **MMD-VQ**—a novel vector quantization approach that achieves direct distributional alignment through Maximum Mean Discrepancy (MMD) (Gretton et al., 2012; Sriperumbudur et al., 2009). Unlike Gaussian-dependent alternatives (Fang et al., 2025), MMD-VQ operates without distributional assumptions, robustly aligning feature and codebook distributions even for complex non-Gaussian data. This fundamental flexibility positions MMD-VQ as an intrinsically compatible solution for VQ-Transplant, particularly advantageous in real-world visual tokenization scenarios where feature distributions frequently diverge from parametric forms.

The method operates on feature vectors $X = \{z_1, z_2, ..., z_N\}$ (spatial features $\boldsymbol{z}_e^{ij}$) and codebook vectors $Y = \{e_1, e_2, ..., e_K\}$, computing the squared MMD distance as:

$$\mathcal{D}_{\text{MMD}}^2(X, Y) = \frac{1}{N^2} \sum_{i=1}^{N} \sum_{j=1}^{N} k(z_i, z_j) + \frac{1}{K^2} \sum_{i=1}^{K} \sum_{j=1}^{K} k(e_i, e_j) - \frac{2}{NK} \sum_{i=1}^{N} \sum_{j=1}^{K} k(z_i, e_j), \quad (12)$$

where $k(\cdot, \cdot)$ denotes a characteristic kernel. Critically, $\mathcal{D}_{\text{MMD}}^2(X, Y) = 0$ iff $\mathcal{P}_X = \mathcal{P}_Y$, establishing MMD as a powerful nonparametric divergence metric. In this implementation, Anonymous (2025) utilize a multi-Gaussian kernel $k(x, y) = \sum_i \exp(-\frac{\|x-y\|^2}{2\sigma_i^2})$ and incorporate $\mathcal{L}_{\text{unique}} = \mathcal{D}_{\text{MMD}}^2(X, Y)$ into Equation 10 to achieve distributional alignment.

Table 4: Reconstruction performance on the ImageNet-1k dataset w.r.t. codebook size.

| Methods | Phase | Tokens | Codebook Size $K$ | $\mathcal{E}(\downarrow)$ | U ($\uparrow$) | PSNR($\uparrow$) | SSIM($\uparrow$) | LPIPS ($\downarrow$) | r-FID($\downarrow$) | r-IS($\uparrow$) |
|---|---|---|---|---|---|---|---|---|---|---|
| MMD VQ | Substitution | 512 | 1024 | 0.318 | 99.6% | 23.75 | 60.8 | 0.162 | 2.68 | 167.6 |
| MMD VQ | Substitution | 512 | 2048 | 0.296 | 99.4% | 24.12 | 62.4 | 0.159 | 2.59 | 168.8 |
| MMD VQ | Substitution | 512 | 4096 | 0.273 | 99.4% | 24.41 | 63.5 | 0.141 | 1.96 | 178.0 |
| MMD VQ | Substitution | 512 | 8192 | 0.252 | 99.5% | 24.67 | 64.6 | 0.135 | 1.85 | 181.0 |
| MMD VQ | Substitution | 512 | 16384 | 0.234 | 99.8% | 24.89 | 65.4 | 0.130 | 1.84 | 183.7 |
| MMD VQ | Substitution | 512 | 32768 | 0.215 | 99.7% | 25.06 | 66.3 | 0.126 | 1.79 | 184.8 |
| MMD VQ | Substitution | 512 | 65536 | 0.201 | 99.9% | 25.24 | 66.8 | 0.121 | 1.69 | 187.3 |
| MMD VQ | Adaptation | 512 | 1024 | 0.318 | 99.6% | 23.06 | 58.8 | 0.148 | 1.90 | 169.8 |
| MMD VQ | Adaptation | 512 | 2048 | 0.296 | 99.4% | 23.58 | 61.2 | 0.137 | 1.63 | 176.6 |
| MMD VQ | Adaptation | 512 | 4096 | 0.273 | 99.4% | 23.89 | 61.8 | 0.128 | 1.28 | 185.1 |
| MMD VQ | Adaptation | 512 | 8192 | 0.252 | 99.5% | 24.11 | 62.9 | 0.121 | 1.18 | 187.9 |
| MMD VQ | Adaptation | 512 | 16384 | 0.234 | 99.8% | 24.31 | 63.7 | 0.115 | 1.05 | 191.2 |
| MMD VQ | Adaptation | 512 | 32768 | 0.216 | 99.9% | 24.53 | 64.7 | 0.110 | 0.97 | 194.1 |
| MMD VQ | Adaptation | 512 | 65536 | 0.201 | 99.9% | 24.65 | 65.0 | 0.106 | 0.86 | 197.1 |

Table 5: Reconstruction performance on the ImageNet-1k dataset w.r.t. token counts.

| Methods | Phase | Tokens | Codebook Size $K$ | $\mathcal{E}(\downarrow)$ | U ($\uparrow$) | PSNR($\uparrow$) | SSIM($\uparrow$) | LPIPS ($\downarrow$) | r-FID($\downarrow$) | r-IS($\uparrow$) |
|---|---|---|---|---|---|---|---|---|---|---|
| MMD VQ | Substitution | 256 | 16384 | 0.369 | 99.6% | 22.97 | 57.2 | 0.194 | 4.91 | 141.4 |
| MMD VQ | Substitution | 512 | 16384 | 0.234 | 99.8% | 24.89 | 65.4 | 0.130 | 1.84 | 183.7 |
| MMD VQ | Substitution | 1024 | 16384 | 0.098 | 100% | 26.40 | 71.0 | 0.100 | 2.01 | 191.7 |
| MMD VQ | Substitution | 2048 | 16384 | 0.035 | 100% | 27.16 | 73.1 | 0.089 | 2.36 | 192.2 |
| MMD VQ | Adaptation | 256 | 16384 | 0.369 | 99.6% | 22.41 | 55.9 | 0.171 | 3.06 | 148.9 |
| MMD VQ | Adaptation | 512 | 16384 | 0.234 | 99.8% | 24.31 | 63.7 | 0.115 | 1.05 | 191.2 |
| MMD VQ | Adaptation | 1024 | 16384 | 0.098 | 100% | 26.03 | 69.6 | 0.079 | 0.54 | 210.1 |
| MMD VQ | Adaptation | 2048 | 16384 | 0.035 | 100% | 27.31 | 73.2 | 0.060 | 0.42 | 217.0 |

Table 6: Equivalence between token count and codebook size

| Methods | Phase | Tokens | Codebook Size $K$ | $\mathcal{E}(\downarrow)$ | U ($\uparrow$) | PSNR($\uparrow$) | SSIM($\uparrow$) | LPIPS ($\downarrow$) | r-FID($\downarrow$) | r-IS($\uparrow$) |
|---|---|---|---|---|---|---|---|---|---|---|
| MMD VQ | Substitution | 512 | $128 \times 128$ | 0.234 | 99.8% | 24.89 | 65.4 | 0.130 | 1.84 | 183.7 |
| MMD VQ | Adaptation | 512 | $128 \times 128$ | 0.234 | 99.8% | 24.31 | 63.7 | 0.115 | 1.05 | 191.2 |
| MMD VQ | Substitution | $512 \times 2$ | 128 | 0.243 | 100% | 24.72 | 64.9 | 0.132 | 1.80 | 184.5 |
| MMD VQ | Adaptation | $512 \times 2$ | 128 | 0.243 | 100% | 24.01 | 63.0 | 0.118 | 1.10 | 190.4 |
| MMD VQ | Substitution | 512 | $256 \times 256$ | 0.201 | 99.9% | 25.24 | 66.8 | 0.121 | 1.69 | 187.3 |
| MMD VQ | Adaptation | 512 | $256 \times 256$ | 0.201 | 99.9% | 24.65 | 65.0 | 0.106 | 0.86 | 197.1 |
| MMD VQ | Substitution | $512 \times 2$ | 256 | 0.212 | 100% | 25.11 | 66.4 | 0.124 | 1.67 | 187.6 |
| MMD VQ | Adaptation | $512 \times 2$ | 256 | 0.212 | 100% | 24.51 | 64.6 | 0.108 | 0.96 | 195.3 |

## G  EXPERIMENTAL DETAILS

**Data Augmentation.**    All experiments were conducted on the ImageNet-1k dataset (Deng et al., 2009). Following Llama Gen (Sun et al., 2024), images were resized to 256×256 resolution using iterative box downsampling.

**Encoder-Decoder Architecture.**    All experiments utilize the VQ-Transplant framework, initialized with a pre-trained VAR tokenizer (Tian et al., 2024), resulting in an encoder-decoder architecture identical to VAR. The encoder—a U-Net (Ronneberger et al., 2015)—downsamples input images by a factor of 16, producing latent features $z_e = \mathcal{E}_\theta(x) \in \mathbb{R}^{16 \times 16 \times 32}$ with $16 \times 16$ spatial resolution.

**Training Details.**    Following (Anonymous, 2025), all experiments were conducted on two NVIDIA H100 GPUs using the AdamW optimizer (Loshchilov & Hutter, 2019) with $\beta_1 = 0.9$ and $\beta_2 = 0.95$. For the VQ module substitution phase, we used an initial learning rate of $10^{-4}$ with linear decay to $10^{-5}$ and trained for 2 epochs. For decoder adaptation, the learning rate was kept constant at $10^{-5}$ and trained for 5 epochs.

**Loss Weight.**    For all experiments, $\lambda_P$ is fixed to 1. In multi-scale quantization experiments, $\lambda_G = 0.5$, while in fixed-scale quantization experiments, $\lambda_G = 0.4$. We set $\gamma = 0.2$ for configurations employing Wasserstein distance (Wasserstein VQ and Wasserstein VP2) and $\gamma = 0.5$ for for configurations using MMD distance (MMD VQ and MMD VP2).

**VQ Implementation.**    We implement five VQ variants within the VQ-Transplant framework: Vanilla VQ (van den Oord et al., 2017), EMA VQ (Razavi et al., 2019), Online VQ (Zheng & Vedaldi, 2023), Wasserstein VQ (Fang et al., 2025), and MMD VQ (Anonymous, 2025). Specifically, we use a pretrained VAR encoder to extract latent feature embeddings $z_e = \mathcal{E}_\theta(x) \in \mathbb{R}^{16 \times 16 \times 32}$. Three CNN layers are then applied to $z_e$ while preserving the output feature dimension. Then, we implement a parallel quantization system where these 32-dimensional feature vectors are partitioned into two 16-

dimensional sub-vectors. Each sub-vector is independently quantized through separate VQ modules before being concatenated to reconstruct the 32-dimensional vectors, as depicted in Figure 4. Finally, three additional CNN layers process the concatenated quantized features to generate the decoder input. Notably The codebook size $K$ is set to 65536 for all methods for fair comparison.

**PQ Implementation.** We implement five PQ variants within the VQ-Transplant framework: Vanilla VP2 (van den Oord et al., 2017), EMA VP2 (Razavi et al., 2019), Online VP2 (Zheng & Vedaldi, 2023), Wasserstein VP2 (Fang et al., 2025), and MMD

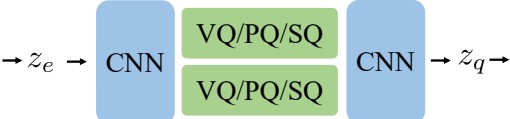

Figure 4: The illustration of implementation details.

VP2 (Anonymous, 2025). Implementation details remain largely identical to standard VQ with one key modification: we set $M = 2$ and further partition the 16-dimensional latent vectors into two 8-dimensional sub-vectors. Each 8-dimensional sub-vector undergoes independent quantization through separate VQ modules with subcodebook sizes of 256. This configuration yields a global codebook size of 65,536, maintaining equivalence with conventional VQ implementations.

**SQ Implementation.** We implement three SQ variants within the VQ-Transplant framework: FSQ (Mentzer et al., 2024a), LFQ (Yu et al., 2024), and BSQ (Zhao et al., 2025). For LFQ implementation, the configuration remains identical to standard VQ, except we substitute the quantization module with LFQ. For BSQ implementation, we apply normalization to each 16-dimensional vector and replace the VQ module with BSQ quantization. FSQ implementation differs in two key aspects: (i) A 3-layer CNN first reduces the latent dimension from 32 to 16, while a subsequent 3-layer CNN expands it back to 32 dimensions, maintaining identical dimensionality between $\boldsymbol{z}_e$ and $\boldsymbol{z}_q$. (ii) Each dimension of the feature vector is discretized into 4 fixed scalar values. Notably, all three variants maintain a global codebook size $K = 65,536$, identical to VQ baselines. This corresponds to $2^{16}$ for LFQ and BSQ, and $4^8$ for FSQ, ensuring fair comparison.

**Remark.** For all quantization algorithms, we uniformly configure two key parameters: (i) Codebook size $K = 65,536$; (ii) Token counts $256 \times 2$ (parallel quantization system). This configuration satisfies Condition 2. Notably, while we make no distributional guarantees regarding quantization inputs, we maintain nearly identical CNN network capacity across all methods except FSQ. Crucially, the encoder output $\boldsymbol{z}_e$ remains identical for all algorithms during quantization. This design ensures: (i) Unbiased evaluation of intrinsic quantization effectiveness; (ii) Satisfaction of Condition 1.

# H ANALYSIS ON CODEBOOK SIZE AND TOKEN COUNT

To investigate the impact of codebook size on quantization performance and reconstruction performance, we incrementally scale the codebook size by a factor of 2, ranging from 1024 to 65536. As presented in Table 4, during the VQ module substitution phase, increasing the codebook size led to a reduction in quantization error $\mathcal{E}$ from 0.318 to 0.201. Furthermore, following decoder adaptation, the key reconstruction metric, r-FID, improved from 1.90 to 0.86. These results demonstrate that codebook size exerts a moderate yet significant influence on both $\mathcal{E}$ and r-FID.

We also examine the effect of token count on quantization performance and reconstruction performance, doubling the token count from 256 to 1024. As observed in Table 5, the token count exhibits a considerably more substantial effect on these metrics. When the token count increases from 256 to 1024, quantization error decreases markedly from 0.369 to 0.035, while r-FID improves substantially from 3.06 to 0.42.

To further investigate the relationship between token count and codebook size, we compared two scenarios: doubling the token count versus squaring the codebook size, as shown in Table 6. These two scenarios exhibited nearly identical performance, which provides empirical support for the equivalence relationship predicted in Section 4.1, specifically in Finding 1: doubling the token count is equivalent to squaring the codebook size in terms of information capacity.

## I    LIMITATIONS

A key limitation of our work is that the linear scaling between quantization error and latent feature distribution variance in Section 4.3 was validated empirically rather than proven theoretically. We hope that a proof for this relationship can be provided in future work.

## J    EXPLANATIONS OF INFORMATION QUANTITY DEFINITIONS IN SECTION 4.1

The quantities $\mathcal{Q}_i$, $\mathcal{Q}_o$, and $\mathcal{Q}_r$ represent the number of bits to encode the input and output, as well as their ratio. These definitions are inspired by the classical Shannon entropy formulation, using bits as the unit of information. It is important to emphasize that Shannon entropy characterizes the **average information content** and therefore depends on the underlying probability distribution. In contrast, our formulation focuses on the **maximum information content**, which does not require specifying or assuming any distribution.

**Definition of $\mathcal{Q}_i$.**  We first define $\mathcal{Q}_i$ based on the above-described entropies, providing explicit expressions for continuous distributions. For the input $z_e \in \mathbb{R}^{h \times w \times d}$, one could consider the continuous Shannon differential entropy $\mathcal{H}(X)$, defined as:

$$\mathcal{H}(X) := -\int p(x) \log_2 p(x) \, dx. \tag{13}$$

If $X \sim \mathcal{N}(\mu, \Sigma)$, the Shannon entropy has the closed-form expression:

$$\mathcal{H}(X) = \frac{1}{2} \log_2 \left[ (2\pi e)^d \det(\Sigma) \right]. \tag{14}$$

However, the true distribution of the spatial features $z_e^{ij} \in \mathbb{R}^d$ is unknown. For moderately high-dimensional features, accurately estimating the feature density is extremely challenging. Even if the density were known, computing the Shannon entropy for non-Gaussian distributions remains difficult. In practice, the continuous information is stored as floating point numbers, and the discrete Shannon entropy serves as an approximation to the differential entropy. The information quantity $\mathcal{Q}_i$ is the maximal value of Shannon entropy in the input space under the 32-bit floating point representation, which is defined as in **Definition 1**.

**Definition of $\mathcal{Q}_o$.**  Similarly, $\mathcal{Q}_o$ is defined based on the above-described entropies, with explicit expressions provided for both cases. It is important to emphasize that $\mathcal{Q}_o$ is defined based on discrete tokens rather than quantized features $z_q$. First, the discrete tokens $r^{ij}$ serve as the input to the subsequent generative model; second, they contain less information than $z_q$ and can be directly mapped to $z_q$ through the codebook.

To express $r^{ij}$ in bits, we use the discrete version of Shannon entropy $\mathcal{H}(X)$:

$$\mathcal{H}(X) := -\sum_x p(x) \log_2 p(x). \tag{15}$$

Accurately computing this entropy requires knowledge of the usage probability $p(x)$ of each code vector in the codebook. In this work, we take an upper bound for the entropy, which is achieved by uniform distribution i.e., $p(x) = 1/K$. Under this assumption, the entropy reduces to:

$$\mathcal{H}(X) = -\sum_{i=1}^{K} \frac{1}{K} \log_2 \frac{1}{K} = \log_2 K. \tag{16}$$

**Definition of $\mathcal{Q}_r$.**  The compression ratio $\mathcal{Q}_r$ is defined as the ratio between the input and output information in the compression system, reflecting the degree of information reduction.

Finally, we emphasize that, although our definitions differ slightly from standard Shannon entropy to ensure computational tractability, the use of maximum information bits does not significantly affect the key insights or conclusions of the paper. These definitions primarily highlight the potential influence of codebook size and token count on the quantization algorithm. For fair comparisons, as stated in **Condition 2**: Compression ratios must be held constant across all algorithms by using identical token counts and codebook sizes.

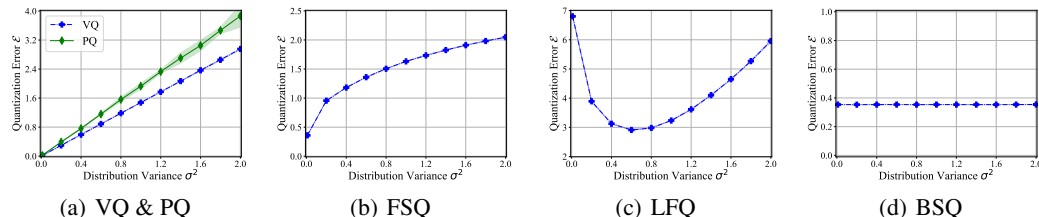

|               |               |               |               |
| :-----------: | :-----------: | :-----------: | :-----------: |
| (a) VQ & PQ   | (b) FSQ       | (c) LFQ       | (d) BSQ       |

Figure 5: Relationship between quantization error and latent feature variance across five quantization algorithms.

# K  QUANTIZATION ERROR VS. LATENT FEATURE VARIANCE: A COMPARATIVE STUDY

To study how quantization error depends on latent variance, we sample $d$-dimensional feature vectors $\{z_i\}_{i=1}^N \sim \mathcal{N}_d(0, \sigma^2 I)$, varying $\sigma^2 \in \{0.01, 0.2, 0.4, 0.6, 0.8, 1.0, 1.2, 1.4, 1.6, 1.8, 2.0\}$ with $d = 8$ and $N = 100{,}000$. Using the same sampled set for all methods, we compute the quantization error separately for the VQ, PQ and SQ variants described below. Each synthetic experiment is repeated five times and the reported curves are averages across trials.

## K.1  LINEARITY IN VQ AND PQ

**VQ.**  Following prior work (Fang et al., 2025; Graf & Luschgy, 2000), VQ is near-optimal when the codebook distribution matches the feature distribution, which minimizes quantization error and maximizes codebook utilization. Motivated by this principle, we construct a near-optimal VQ solution by sampling code vectors from $\mathcal{N}_d(0, \sigma^2 I)$. For each value of $\sigma^2$, we use a codebook of size $K = 4096$ and assign each feature vector to its nearest code vector to compute the quantization error. Each experiment is repeated five times, and the averaged results are shown in Figure 5(a), which clearly demonstrates a pronounced linear relationship between quantization error and latent distribution variance.

**PQ.**  For PQ we split each feature into two 4-dimensional sub-vectors and quantize each subspace using VQ with sub-codebook size 64. By sampling 64 code vectors per subspace (distribution-matching), the overall PQ codebook size becomes $K = 64^2 = 4096$, matching the VQ setup. The averaged PQ results are also shown in Figure 5(a). PQ exhibits an approximately linear dependence of quantization error on latent variance, with small deviations introduced by the product-quantization decomposition.

**Interpretation.**  Both VQ and PQ display approximate linear scaling because codebooks sampled to match a Gaussian source scale with the source variance: the expected nearest-neighbor distance (hence the quantization error) grows proportionally with $\sigma^2$. PQ preserves this scaling within each subspace, so summing subspace errors yields an overall trend that remains approximately linear. Under the fair setting (identical codebook size and identical sampled feature vectors), PQ consistently produces larger quantization error than VQ for every tested $\sigma^2$, which aligns with our theoretical result in Section 4.4 that the optimal VQ solution outperforms PQ.

## K.2  LACK OF LINEARITY IN SQ

To date, no prior work has proposed optimal SQ algorithms. Among the three classical SQ algorithms—FSQ (Mentzer et al., 2024a), LFQ (Yu et al., 2024), and BSQ (Zhao et al., 2025)—all still rely on fixed codebooks. Such fixed strategies are inherently suboptimal. As shown in Figures 5(b), 5(c), and 5(d), all three algorithms exhibit a pronounced non-linear relationship between quantization error and the variance of the latent distribution. We now analyze each method in detail.

**FSQ.**  FSQ rescales each latent dimension via $z_i \mapsto \lfloor \frac{K_2}{2} \rfloor \tanh(z_i)$ and then quantizes it using $K_2$ discrete integers. In our experiments, we set $K_2 = 4$, resulting in an overall codebook size of $4^8 = 65{,}536$. As noted above, this fixed codebook strategy is suboptimal because it cannot adapt to the distribution of the latent features, limiting its ability to minimize quantization error. This

limitation is reflected in the characteristic curve of quantization error versus latent feature variance shown in Figure 5(b).

**LFQ.** LFQ does not rescale the latent distribution and uses a fixed set of discrete integers $\{-1, 1\}$ for quantization. The overall codebook size is therefore $2^8 = 256$. In this setting, when the latent variance $\sigma^2$ is very small or very large, the quantization error increases, while a moderate variance (e.g., $\sigma^2 = 0.6$) produces lower quantization error, as illustrated in Figure 5(c).

**BSQ.** BSQ normalizes each latent feature vector to lie on the unit hypersphere via $z_i \mapsto \frac{z_i}{\|z_i\|_2} \in \mathbb{S}^{d-1}$ Mathematically, if $Z \sim \mathcal{N}_d(0, \sigma^2 I)$, then $Y := Z/\|Z\|_2$ is uniformly distributed on the unit hypersphere $\mathbb{S}^{d-1}$. Consequently, after normalization, feature vectors with different variances $\sigma^2$ all follow the same distribution. This explains why BSQ exhibits a nearly constant relationship between quantization error and latent feature variance, as shown in Figure 5(d). Notably, BSQ always uses $K_2 = 2$, resulting in an overall codebook size of $2^8 = 256$.

## L  JOINT OPTIMIZATION OF ENCODER, DECODDR, AND QUANTIZATION MODULES

In the original Stage II in Appendix E, only the decoder $\mathcal{D}_\varphi$ is updated while keeping the pretrained encoder $\mathcal{E}_{\theta^*}$ and newly trained VQ module $\mathcal{Q}_{\phi^*}^{\text{new}}$ frozen. This approach addresses the mismatch between the updated quantized latent space and the frozen decoder, but it does not allow the encoder to adapt to the new quantization or jointly refine the reconstruction capability.

As an alternative, we also introduce a joint optimization scheme in Stage II, where the encoder, decoder, and VQ module are updated simultaneously. Let $z_e = \mathcal{E}_\theta(x)$ denote the encoder's latent embedding, and $z_q = \mathcal{Q}_\phi(ze)$ denote the quantized latent from the VQ module. The decoder reconstructs the input as $\hat{x} = \mathcal{D}_\varphi(z_q)$. The overall joint optimization objective integrates both the VQ reconstruction loss and the decoder reconstruction loss:

$$\mathcal{L}_{\text{Joint}}(\theta, \phi, \varphi) = \|\text{sg}(z_e) - z_q\|_2^2 + \beta\|z_e - \text{sg}(z_q)\|_2^2 + \gamma \mathcal{L}_{\text{unique}}(\mathcal{Q}_\phi^{\text{new}}),$$
$$+ \|\hat{x} - x\|_2^2 + \lambda_P \mathcal{L}_{\text{Per}} + \lambda_G \mathcal{L}_{\text{GAN}},$$

where $\mathcal{L}_{\text{unique}}$ enforces codebook uniqueness (e.g., Wasserstein loss for Wasserstein VQ (Fang et al., 2025) or MMD loss for MMD VQ (Anonymous, 2025)), and $\mathcal{L}_{\text{Per}}$ and $\mathcal{L}_{\text{GAN}}$ correspond to perceptual and adversarial losses that promote high-quality reconstruction. The parameter $\beta$ is fixed to 0.25, while $\gamma$, $\lambda_P$, and $\lambda_G$ are hyperparameters balancing the respective terms, as detailed in Appendix G.

In this setup, all three components—encoder $\mathcal{E}_\theta$, decoder $\mathcal{D}_\varphi$, and VQ module $\mathcal{Q}_\phi$, are updated jointly. To initialize the training, we load all parameters from Stage I, ensuring that the encoder, decoder, and VQ module start from the previously optimized representations. Joint optimization enables the encoder to adapt to the updated quantized space, allows the VQ module to refine the codebook representations, and improves the decoder's ability to reconstruct images accurately from the newly quantized latent features. For adversarial training, we follow prior works (Tian et al., 2024; Chen et al., 2025; Li et al., 2025) and employ a frozen DINO-S (Caron et al., 2021; Oquab et al., 2024) discriminator with a StyleGAN-like architecture (Karras et al., 2020; 2019), augmented with DiffAug (Zhao et al., 2020), consistency regularization (Zhang et al., 2019), and LeCAM regularization (Tseng et al., 2021).

We conduct experiments on ImageNet-1K to compare the decoder-only and joint-optimization strategies, as summarized in Table 7. Compared with the decoder-only training scheme, we observe that joint optimization consistently improves the reconstruction performance of most quantization algorithms. This benefit is expected because allowing the encoder, quantizer, and decoder to co-adapt enables the model to learn feature representations that are inherently more compatible with the quantization constraints, thereby reducing mismatch and improving end-to-end reconstruction fidelity.

Notably, when the encoder is not frozen, the raw quantization error $\mathcal{E}$ becomes less indicative of the actual reconstruction quality. This occurs because different quantization algorithms learn feature distributions with substantially different variances, and as shown in Section 4.3, the quantization error

Table 7: Comparative reconstruction performance of VQ, PQ, and SQ quantization methods on ImageNet-1K. [†]: Results cited from VQ-Transplant (Anonymous, 2025). Within each quantization type and strategy, optimal values are underlined; overall best results per metric are **bold**.

| Approaches | Types | Training Strategies | Tokens | $K$ | $\mathcal{E}(\downarrow)$ | U ($\uparrow$) | PSNR($\uparrow$) | SSIM($\uparrow$) | LPIPS ($\downarrow$) | r-FID($\downarrow$) | r-IS($\uparrow$) |
|---|---|---|---|---|---|---|---|---|---|---|---|
| Vanilla VQ[†] | VQ | Decoder-Only | 512 | 65536 | 0.422 | 0.2% | 21.19 | 50.7 | 0.209 | 5.05 | 118.9 |
| EMA VQ[†] | VQ | Decoder-Only | 512 | 65536 | 0.217 | 65.5% | 24.36 | 64.1 | 0.111 | 0.99 | 194.3 |
| Online VQ[†] | VQ | Decoder-Only | 512 | 65536 | 0.280 | 13.5% | 23.84 | 61.6 | 0.130 | 1.38 | 182.9 |
| Wasserstein VQ[†] | VQ | Decoder-Only | 512 | 65536 | **0.201** | 99.6% | **24.68** | **65.4** | **0.106** | 0.92 | 195.5 |
| MMD VQ[†] | VQ | Decoder-Only | 512 | 65536 | **0.201** | 99.9% | 24.65 | 65.0 | **0.106** | **0.86** | **197.1** |
| Vanilla VP2 | PQ | Decoder-Only | 512 | 65536 | 0.233 | 59.2% | 24.28 | 64.0 | 0.114 | 1.07 | 191.7 |
| EMA VP2 | PQ | Decoder-Only | 512 | 65536 | 0.209 | **100%** | 24.55 | 64.9 | 0.107 | 0.93 | 195.4 |
| Online VP2 | PQ | Decoder-Only | 512 | 65536 | 0.211 | **100%** | 24.53 | 64.7 | 0.108 | 0.95 | 195.3 |
| Wasserstein VP2 | PQ | Decoder-Only | 512 | 65536 | 0.217 | **100%** | 24.44 | 64.6 | 0.110 | 0.99 | 193.5 |
| MMD VP2 | PQ | Decoder-Only | 512 | 65536 | 0.212 | **100%** | 24.43 | 64.5 | 0.109 | 0.95 | 196.1 |
| FSQ | SQ | Decoder-Only | 512 | 65536 | 0.300 | 71.0% | 23.27 | 59.1 | 0.134 | 1.52 | 179.3 |
| LFQ | SQ | Decoder-Only | 512 | 65536 | 0.279 | 29.8% | 23.42 | 60.7 | 0.130 | 1.30 | 183.2 |
| BSQ | SQ | Decoder-Only | 512 | 65536 | 0.231 | **100%** | 24.06 | 63.6 | 0.117 | 1.07 | 190.8 |
| Vanilla VQ | VQ | Joint Optimization | 512 | 65536 | 0.173 | 0.2% | 22.10 | 53.7 | 0.164 | 2.30 | 161.5 |
| EMA VQ | VQ | Joint Optimization | 512 | 65536 | 0.077 | 91.9% | 24.52 | 65.3 | 0.103 | 0.87 | 197.1 |
| Online VQ | VQ | Joint Optimization | 512 | 65536 | 0.094 | 16.7% | 24.10 | 62.1 | 0.120 | 1.19 | 188.5 |
| Wasserstein VQ | VQ | Joint Optimization | 512 | 65536 | 0.162 | 99.8% | 24.66 | 65.8 | 0.102 | 0.79 | 200.6 |
| MMD VQ | VQ | Joint Optimization | 512 | 65536 | 0.235 | 99.8% | **24.88** | **66.2** | **0.100** | **0.74** | **201.8** |
| Vanilla VP2 | PQ | Joint Optimization | 512 | 65536 | 0.067 | 92.0% | 24.53 | 64.4 | 0.106 | 0.92 | 197.6 |
| EMA VP2 | PQ | Joint Optimization | 512 | 65536 | 0.069 | **100%** | 24.67 | 65.3 | 0.102 | 0.81 | 199.6 |
| Online VP2 | PQ | Joint Optimization | 512 | 65536 | 0.070 | **100%** | 24.63 | 64.6 | 0.104 | 0.87 | 198.4 |
| Wasserstein VP2 | PQ | Joint Optimization | 512 | 65536 | 0.176 | **100%** | 24.57 | 64.9 | 0.102 | 0.88 | 198.7 |
| MMD VP2 | PQ | Joint Optimization | 512 | 65536 | 0.178 | **100%** | 24.78 | 65.4 | 0.102 | 0.80 | 200.9 |
| FSQ | SQ | Joint Optimization | 512 | 65536 | 0.209 | 30.7% | 22.66 | 56.6 | 0.143 | 1.53 | 179.8 |
| LFQ | SQ | Joint Optimization | 512 | 65536 | 0.344 | 9.4% | 22.95 | 58.0 | 0.137 | 1.38 | 180.7 |
| BSQ | SQ | Joint Optimization | 512 | 65536 | 0.193 | **100%** | 24.06 | 63.3 | 0.113 | 0.95 | 194.4 |

scales linearly with the latent feature variance. As a result, $\mathcal{E}$ no longer provides a fair comparison across algorithms in the joint-optimization setting.

Even under these conditions, we can still clearly identify the best-performing quantization methods. For example, MMD-VQ consistently outperforms all PQ-based methods, and within the PQ family, MMD-VP2 achieves the best results, surpassing all SQ algorithms. These empirical findings are fully aligned with our theoretical analysis in Section 4.4.

