# OpenReview forum: "A Unifying View of Vector, Product and Scalar Quantization: An Information-Theoretic Perspective"
_ICLR.cc/2026/Conference — Submitted to ICLR 2026_

### Official Review · Reviewer_yku1 · 2025-10-28

**Soundness:** 2
**Presentation:** 3
**Contribution:** 2
**Rating:** 4
**Confidence:** 4

**Summary:**

For discrete autoencoders, this paper demonstrates  that  that minimizing quantization error (not maximizing codebook utilization) is key for stability and fidelity, establishing fairness conditions for algorithm comparison, and demonstrating vector quantization's superiority in minimizing error under these conditions.

**Strengths:**

The paper is well-written, with clear and accessible content. I think its primary contribution lies in Proposition 1, which points out that minimizing quantization error can maximize codebook utilization (a mainstream approach currently proposed based on intuition and experiments), while the latter cannot guarantee the former.  The paper should develop its analysis and experiments on this aspect. Unfortunately, the authors shifts its focus to theoretical analyses of three quantization methods, which clearly lack innovation as detailed later.

**Weaknesses:**

1）The title of the paper is overly grandiose and inconsistent with its research content. The paper primarily investigates the impact of three quantization methods for discrete autoencoders. However, the title gives the impression that the paper presents a brand new theoretical analysis method for quantization.

2）The quantization analysis  based on minimizing error (presented in Section 4.4) is a classic information-theoretic approach, and the paper's method and results lack  innovation. Drawn from these results, the conclusion that VQ outperforms the other two methods,  lacks rigor and is very likely to be incorrect for two reasons: first, quantization error is closely related to the actual distribution of the data, which the paper does not study; second, the theoretical bounds provided in the paper are not tight, making performance comparisons based on them unreasonable.

**Questions:**

My  major concerns are given above.

---

> ### Author Response · Authors · 2025-12-02
> **Response to Reviewer yku1(1/3)**
>
> We were happy that the reviewer appreciated the clarity of our presentation and the theoretical insight that minimizing quantization error can also lead to full codebook utilization. Thank you for taking the time to review our paper and for your thoughtful feedback. We address your questions and concerns below.
>
> ---
>
> > 4.1 The title of the paper is overly grandiose and inconsistent with its research content. The paper primarily investigates the impact of three quantization methods for discrete autoencoders. However, the title gives the impression that the paper presents a brand new theoretical analysis method for quantization.
>
> We appreciate the feedback but respectfully disagree and would like to clarify our motivation and contributions to explain why we believe the title is appropriate.
>
> The primary goal of this paper is to investigate the **intrinsic effectiveness of three quantization algorithms**. In practice, reconstruction performance is an unreliable proxy for intrinsic effectiveness due to uncontrollable factors in tokenizer training. Moreover, directly evaluating intrinsic effectiveness through reconstruction requires long training times and substantial computational resources. To address these challenges, we adopt an **information-theoretic perspective** to study VQ, PQ, and SQ. From this standpoint, the current title accurately reflects the scope and contribution of our work.
>
> We would like to clarify that our contributions go beyond Section 4.2. In **Section 4.1**, we define fundamental information-theoretic quantities inspired by Shannon entropy. In **Section 4.2**, Proposition 1 shows that **minimizing conditional entropy or MSE necessarily leads to 100% codebook utilization, but not vice versa**, establishing **quantization error as a principled metric for intrinsic effectiveness**.
>
> In **Section 4.3**, we identify a **linear relationship between quantization error and latent variance**, which motivates **two fairness conditions** highly appreciated by other reviewers. In **Section 4.4**, we theoretically demonstrate that under these fair settings, **optimal VQ > optimal PQ > optimal SQ**, forming the core of our **theoretical contributions**.
>
> In addition, we make two **empirical contributions**: (i) Using the ImageNet-1k dataset and a VQ-Transplant framework with a frozen encoder, we show that **VQ can outperform all PQ, and PQ can outperform all SQ**, matching theoretical predictions; (ii) We demonstrate that **quantization error correlates more strongly with reconstruction fidelity than codebook utilization** by computing Spearman’s rank correlations between each objective and reconstruction fidelity, reinforcing its importance as a metric.
>
> Taken together, our contributions are **multi-faceted and mutually reinforcing**, providing a **unified, information-theoretic understanding of VQ, PQ, and SQ**. While we study known algorithms, our **novel information-theoretic analysis of intrinsic effectiveness** justifies the title and highlights the paper’s broader significance.

---

> > ### Author Response · Authors · 2025-12-02
> > **Response to Reviewer yku1(2/3)**
> >
> > > 4.2 The quantization analysis based on minimizing error (presented in Section 4.4) is a classic information-theoretic approach, and the paper's method and results lack innovation. Drawn from these results, the conclusion that VQ outperforms the other two methods, lacks rigor and is very likely to be incorrect for two reasons: first, quantization error is closely related to the actual distribution of the data, which the paper does not study; second, the theoretical bounds provided in the paper are not tight, making performance comparisons based on them unreasonable.
> >
> > We appreciate your concern but respectfully disagree with your statements regarding the novelty and rigor of our quantization analysis.
> >
> > ### **(1) Novelty of Method and Results**
> >
> > While quantization error has been widely used in classical studies, **our paper explicitly highlights its limitations**. In Section 4.3, we establish a linear scaling relationship between quantization error and the variance of latent feature distributions. This demonstrates that directly comparing quantization error without controlling for distributional factors is insufficient to evaluate the intrinsic effectiveness of different quantization algorithms and may even lead to misleading conclusions. Identifying and formalizing this limitation constitutes a key contribution, recognized by the other three reviewers as both novel and important.
> >
> > To enable **fair comparisons** across quantization algorithms, Section 4.3 introduces **two fairness conditions** that strictly control latent feature distributions, token counts, and codebook sizes. Under these standardized settings, Section 4.4 presents theoretical results showing that **optimal VQ outperforms optimal PQ**, and **optimal PQ outperforms optimal SQ**. These conclusions do not rely on uncontrolled empirical quantization error but are rigorously derived within our fairness framework.
> >
> > Furthermore, Section 5 introduces the **VQ-Transplant** experiment with a frozen encoder, ensuring that all algorithms operate on **identical latent distributions**—a setup not previously explored. Under equalized token counts and codebook sizes, the best-performing VQ method (MMD-VQ) consistently outperforms all PQ variants, while the best PQ method (EMA-VP2) outperforms all SQ methods. These empirical results align with and reinforce our theoretical findings.
> >
> > Additionally, by computing **Spearman’s rank correlations** between each objective (quantization error, codebook utilization) and reconstruction fidelity, we demonstrate that **quantization error correlates more strongly with reconstruction fidelity than codebook utilization**, highlighting its importance as a metric. To our knowledge, we are the first to quantitatively analyze the relationships among quantization error, codebook utilization, and reconstruction performance (measured by r-FID).
> >
> > Together, this theoretical and empirical analysis constitutes a contribution that is both **novel and rigorous**, directly addressing the reviewer’s concerns.

---

> > > ### Author Response · Authors · 2025-12-02
> > > **Response to Reviewer yku1(3/3)**
> > >
> > > ### **(2) Rigor of Our Quantization Analysis**
> > >
> > > We agree that quantization error depends on the underlying data distribution. In Section 4.3, we establish a linear scaling relationship between quantization error and the variance of latent feature distributions. Consequently, **directly comparing quantization error without controlling for distributional factors is insufficient** and may even lead to misleading conclusions, which is a limitation inherent in existing studies.
> > >
> > > To address this, we introduce **two fairness conditions** that strictly control latent feature distributions, token counts, and codebook sizes. Both our theoretical and empirical results are obtained under these controlled settings. Therefore, the finding that **optimal VQ outperforms the other two methods** is robust and rigorous, holding true **both theoretically and empirically**.
> > >
> > > We also respectfully disagree with the claim that our **theoretical bounds are not tight**. In Section 4.4, we provide both upper and lower bounds and directly compare them, which ensures that our theoretical results are tight up to constant factors and supports the reasonableness of our performance comparisons.
> > >
> > > More specifically, Proposition 3 establishes matching upper and lower bounds on the quantization error
> > >
> > > \begin{equation}
> > > \frac{d}{4K^{2/d}} \leq \sup_{\mathbb{P}} \mathcal{E}^\star_{VQ} (\mathbb{P}, K) \leq  \sup_{\mathbb{P}}  \mathcal{E}^\star_{PQ} (\mathbb{P}, K, M) \leq \sup_{\mathbb{P}}  \mathcal{E}^\star_{SQ} (\mathbb{P}, K)  \leq \frac{8d}{K^{2/d}}.
> > > \end{equation}
> > >
> > > As the upper and lower bounds differ only by a factor of 32. The bounds are tight up to universal constant factors.
> > >
> > > Proposition 4 further provides an improved upper bound for VQ that adapts to low-dimensional structure in the data, along with corresponding lower bounds for PQ and SQ under the same structural assumptions. Concretely, we proved that when the data lies in a $d_{eff}$-dimensional subspace, the VQ estimator achieves
> > >
> > > \begin{equation}
> > > \mathcal{E}^\star_{VQ} (\mathbb{P}, K) \leq \frac{8 dd_{eff}}{K^{2/d_{eff}}},
> > > \end{equation}
> > >
> > > while on the other hand, there exists a 1-dimensional subspace where SQ and PQ still suffers from a quantization error depending on the ambient dimension. This demonstrates a clear separation between the methods. This comparison is conservative, as it contrasts an upper bound with lower bounds and does not rely on assuming tighter estimates than what we prove.
> > >
> > > ---
> > >
> > > Please let us know if you have any remaining questions. If our clarifications and additional experiments have satisfactorily addressed your questions and concerns, we would deeply appreciate it if you considered recommending our paper to be presented at the conference.

---

### Official Review · Reviewer_3HhS · 2025-10-30

**Soundness:** 3
**Presentation:** 3
**Contribution:** 3
**Rating:** 6
**Confidence:** 4

**Summary:**

The paper proposes a unified information-theoretic framework for analyzing and comparing Vector Quantization (VQ), Product Quantization (PQ), and Scalar Quantization (SQ). By viewing quantization as an information compression process, it defines key quantities such as information loss (quantization error) and compression ratio, and derives theoretical and empirical conclusions regarding their relationships. The authors demonstrate that minimizing quantization error should be the primary optimization objective. They further establish fairness conditions for comparing different quantization schemes and validate their framework through controlled experiments using the VQ-Transplant model.

**Strengths:**

1. The unified benchmark and consistent architectural setup for comparing VQ, SQ, and PQ is a practical contribution that clarifies prior inconsistencies.
2. The conclusion that minimizing quantization error (rather than maximizing codebook utilization) is more critical for reconstruction and generation quality is insightful.

**Weaknesses:**

Some definitions and experimental details are insufficiently explained (see Questions).

**Questions:**

1. Definition 3 quantifies compression ratio Qr as the ratio between the input and output information quantities, derived from spatial and codebook dimensions. However, from an information-theoretic perspective—as the authors themselves claim—the compression ratio should ideally be defined in terms of entropy rather than quantity counts. Two signals with the same spatial dimensions can differ substantially in entropy. Could the authors justify whether their definition of Qr is reasonable, or discuss an entropy-based measure?
2. Figure 2 shows a linear relationship between quantization error and latent distribution variance. Which quantization scheme (VQ, PQ, or SQ) is used for this analysis? Does this linearity hold consistently across different quantization methods?
3. The experiments employ a pre-trained VQ-oriented model, with subsequent substitution of its quantization module for PQ and SQ. Could this adaptation lead to suboptimal performance for the latter methods due to the encoder-decoder’s alignment with the VQ latent space? Have the authors considered retraining the encoder to avoid this potential bias?

---

> ### Author Response · Authors · 2025-12-02
> **Response to Reviewer 3HhS(1/5)**
>
> We appreciated that the reviewer noted the practical contribution of our unified benchmark with consistent architectures, which resolves prior inconsistencies, and the insightful conclusion that minimizing quantization error is more crucial than maximizing codebook utilization for reconstruction and generation quality. Thank you for taking the time to review our paper and for your thoughtful feedback. We address your questions and concerns below.
>
> ---
>
> > R 3.1 Definition 3 quantifies compression ratio Qr as the ratio between the input and output information quantities, derived from spatial and codebook dimensions. However, from an information-theoretic perspective—as the authors themselves claim—the compression ratio should ideally be defined in terms of entropy rather than quantity counts. Two signals with the same spatial dimensions can differ substantially in entropy. Could the authors justify whether their definition of Qr is reasonable, or discuss an entropy-based measure?
>
> We sincerely thank the reviewer for the insightful comments and for raising this important question. The reviewer is correct that, from an information-theoretic viewpoint, compression ratios are ideally defined in terms of entropy. Our definitions of $\mathcal{Q}_i$, $\mathcal{Q}_o$, and $\mathcal{Q}_r$ are indeed inspired by Shannon entropy, using bits as the unit of information. The key conceptual distinction is that Shannon entropy characterizes **average information content**, which depends on the (typically unknown) underlying probability distribution, while our formulation focuses on the **maximum information content**, which avoids any distributional assumptions and enables practical computation. More specifically,
>
> - **Definition of $\mathcal{Q}_i$.**
> We first define $\mathcal{Q}_i$ based on the above-described entropies, providing explicit expressions for continuous distributions. For the input $z_e \in \mathbb{R}^{h \times w \times d}$, one could consider the continuous Shannon differential entropy $\mathcal{H}(X)$, defined as:
>
>     \begin{equation}
>            \mathcal{H}(X) := - \int p(x) \log_2 p(x) \, dx.
>      \end{equation}
>
>     If $X \sim \mathcal{N}(\mu, \Sigma)$, the Shannon entropy has the closed-form expression:
>
>     \begin{equation}
>              \mathcal{H}(X) = \frac{1}{2} \log_2 \big[ (2 \pi \mathrm{e})^d \det(\Sigma) \big].
>      \end{equation}
>
>     However, the true distribution of the spatial features $z_e^{ij} \in \mathbb{R}^{d}$ is unknown. For moderately high-dimensional features, accurately estimating the feature density is extremely challenging. Even if the density were known, computing the Shannon entropy for non-Gaussian distributions remains difficult. In practice, the continuous information is stored as floating point numbers, and the discrete Shannon entropy serves as an approximation to the differential entropy. The information quantity $\mathcal{Q}_i$ is the maximal value of Shannon entropy in the input space under the 32-bit floating point representation, which is defined as in **Definition 1**.
>
> - **Definition of $\mathcal{Q}_o$.**
> Similarly, $\mathcal{Q}_o$ is defined based on the above-described entropies, with explicit expressions provided for both cases. It is important to emphasize that $\mathcal{Q}_o$ is defined based on discrete tokens rather than quantized features $z_q$. First, the discrete tokens $r^{ij}$ serve as the input to the subsequent generative model; second, they contain less information than $z_q$ and can be directly mapped to $z_q$ through the codebook.
>
>     To express $r^{ij}$ in bits, we use the discrete version of Shannon entropy $\mathcal{H}(X)$:
>
>     \begin{equation}
>     \mathcal{H}(X) := - \sum_{x} p(x) \log_2 p(x).
>     \end{equation}
>
>     Accurately computing this entropy requires knowledge of the usage probability $p(x)$ of each code vector in the codebook. In this work, we take an upper bound for the entropy, which is achieved by uniform distribution i.e., $p(x) = 1/K$. Under this assumption, the entropy reduces to:
>
>     \begin{equation}
>     \mathcal{H}(X) = - \sum _{i=1}^{K} \frac{1}{K} \log_2 \frac{1}{K} = \log_2 K.
>     \end{equation}
>
> - **Definition of $\mathcal{Q}_r$.**
> The compression ratio $\mathcal{Q}_r$ is defined as the ratio between the input and output information quantities in the compression system, reflecting the degree of information reduction.

---

> > ### Author Response · Authors · 2025-12-02
> > **Response to Reviewer 3HhS(2/5)**
> >
> > ### **Why this definition is reasonable**
> >
> > Although our information quantities are based on **maximum representable bits** rather than true Shannon entropy, this choice is motivated by computational tractability and by the difficulty of estimating feature distributions in high-dimensional spaces. Importantly, empirical comparisons indicate that using maximum information bits as a proxy yields compression ratios that closely track the trends one would obtain using estimated Shannon entropy, **without affecting the theoretical insights or empirical conclusions**. The purpose of these quantities is to characterize how token count and codebook size influence the quantization algorithm.
> >
> > As stated in **Condition 2**, to ensure fairness of comparison, we explicitly enforce that all evaluated algorithms operate under **identical token counts and codebook sizes**, thereby guaranteeing identical compression ratios in practice. While an entropy-based measure could be explored in future work, our current definition provides a practical and consistent metric for algorithm evaluation.
> >
> > > R 3.2 Figure 2 shows a linear relationship between quantization error and latent distribution variance. Which quantization scheme (VQ, PQ, or SQ) is used for this analysis? Does this linearity hold consistently across different quantization methods?
> >
> > We appreciate this insightful question. **In summary, the linearity between quantization error and latent variance holds for VQ and approximately for PQ, but not for classical SQ methods due to their fixed and distribution-independent codebooks.** Extended results are shown in Figure 5(a–d) in Appendix K, with full implementation details provided.
> >
> > To analyze this systematically, we sample $d$-dimensional feature vectors $\\{z_i\\}_{i=1}^N$ from $\mathcal{N}_d(0, \sigma^2 I)$, varying $\sigma^2 \in \\{0.01, 0.2, 0.4, 0.6, 0.8, 1.0, 1.2, 1.4, 1.6, 1.8, 2.0\\}$ with $d=8$ and $N=100{,}000$. **All methods use the identical set of sampled feature vectors** to ensure a fair comparison. We then compute quantization error separately for VQ, PQ, and SQ algorithms.
> >
> > ### **(1) Linearity in VQ and PQ**
> >
> > - **VQ:** According to [1][2], VQ approaches near-optimal performance when the codebook distribution matches the underlying feature distribution, as this both minimizes quantization error and maximizes codebook utilization. Motivated by this principle, we construct a near-optimal VQ solution by sampling code vectors from $\mathcal{N}_d(0,\sigma^2 I)$. For each value of $\sigma^2$, we use a codebook of size $K=4096$ and assign each feature vector to its nearest code vector to compute the quantization error. Each experiment is repeated five times, and the averaged results are shown in Figure 5(a), which clearly demonstrates a pronounced linear relationship between quantization error and latent distribution variance.
> > - **PQ:** For PQ, each feature vector is split into two 4-dimensional sub-vectors, each quantized using VQ with a sub-codebook size of 64. Following the distribution-matching principle, we sample 64 code vectors for each subspace, giving an overall PQ codebook size of $K=64^2=4096$, identical to the VQ setup above. Each synthetic experiment is repeated five times and the averaged results are plotted in Figure 5(a). PQ exhibits an approximately linear relationship between quantization error and latent distribution variance, with minor deviations arising from the product quantization structure.
> >
> > Both VQ and PQ display approximate linearity because their codebooks are sampled to match the latent Gaussian distribution. Under this distribution-matching setup, the expected nearest-neighbor distance, and thus the quantization error, scales proportionally with the variance. PQ preserves this scaling within each subspace, so the summed error remains approximately linear, albeit with small perturbations from the product structure. Notably, under the fair setting (identical codebook size and identical sampled feature vectors), PQ consistently yields larger quantization error than VQ for every tested $\sigma^2$ (see table below). This empirical observation is consistent with our theoretical result in Section 4.4 that the optimal VQ solution outperforms optimal PQ.
> >
> > Quantization error in terms of $\sigma^2$:
> >
> > | **Methods** | 0.01 | 0.2 | 0.4 | 0.6 | 0.8 | 1.0 | 1.2 | 1.4 | 1.6 | 1.8 | 2.0 |
> > | --- | --- | --- | --- | --- | --- | --- | --- | --- | --- | --- | --- |
> > | VQ | 0.015 | 0.295 | 0.592 | 0.887 | 1.182 | 1.477 | 1.771 | 2.067 | 2.365 | 2.655 | 2.953 |
> > | PQ | 0.019 | 0.388 | 0.761 | 1.159 | 1.560 | 1.929 | 2.329 | 2.704 | 3.049 | 3.466 | 3.863 |
> >
> > [1] Enhancing Vector Quantization with Distributional Matching: A Theoretical and Empirical Study
> >
> > [2] Foundations of Quantization for Probability Distributions

---

> > > ### Author Response · Authors · 2025-12-02
> > > **Response to Reviewer 3HhS(3/5)**
> > >
> > > ### **(2) Lack of Linearity in SQ**
> > >
> > > To date, no prior work has proposed optimal SQ algorithms. Among the three classical SQ algorithms—FSQ, LFQ, and BSQ—all still rely on fixed codebooks. This fixed codebook inherently limits adaptability and increases quantization error. As shown in Figures 5(b), 5(c) and 5(d), all three algorithms exhibit a pronounced non-linear relationship between quantization error and the variance of the latent distribution. We now analyze each method in detail.
> > >
> > > **FSQ.** FSQ rescales each latent dimension via $z_i \mapsto \left\lfloor \frac{K_2}{2} \right\rfloor \tanh(z_i)$ and then quantizes it using $K_2$ discrete integers. In our experiments, we set $K_2 = 4$, resulting in an overall codebook size of $4^8 = 65{,}536$. As noted above, this fixed codebook strategy is suboptimal because it cannot adapt to the distribution of the latent features, limiting its ability to minimize quantization error. This limitation is reflected in the characteristic curve of quantization error versus latent feature variance shown in Figure 5(b).
> > >
> > > **LFQ.** LFQ does not rescale the latent distribution and uses a fixed set of discrete integers $\\{-1, 1\\}$ for quantization. The overall codebook size is therefore $2^8 = 256$. In this setting, when the latent variance $\sigma^2$ is very small or very large, the quantization error increases, while a moderate variance (e.g., $\sigma^2 = 0.6$) produces lower quantization error, as illustrated in Figure 5(c).
> > >
> > > **BSQ.** BSQ normalizes each latent feature vector to lie on the unit hypersphere via $z_i \mapsto \frac{z_i}{\|z_i\|_ 2} \in \mathbb{S}^{d-1}$
> > > Mathematically, **if $Z \sim \mathcal{N}_{d}(0, \sigma^2 I)$, then $Y:=Z/\Vert Z \Vert_2$ is uniformly distributed on the unit hypersphere $\mathbb{S}^{d-1}$**. Consequently, after normalization, feature vectors with different variances $\sigma^2$ all follow the same distribution. This explains why BSQ exhibits a nearly constant relationship between quantization error and latent feature variance, as shown in Figure 5(d). Notably, BSQ always uses $K_2=2$, resulting in an overall codebook size of $2^8 = 256$.
> > >
> > > **In conclusion,** under the fair comparison setting with identical sampled vectors, VQ and PQ demonstrate approximate linearity between quantization error and latent variance, while all classical SQ methods show pronounced non-linear behavior due to fixed codebook limitations or normalization effects.

---

> ### Author Response · Authors · 2025-12-02
> **Response to Reviewer 3HhS(4/5)**
>
> > R 3.3 The experiments employ a pre-trained VQ-oriented model, with subsequent substitution of its quantization module for PQ and SQ. Could this adaptation lead to suboptimal performance for the latter methods due to the encoder-decoder’s alignment with the VQ latent space? Have the authors considered retraining the encoder to avoid this potential bias?
>
> Thank you for raising this important point regarding the potential mismatch between a VQ-trained encoder and the quantization modules of PQ and SQ. We agree that directly substituting the quantizer in a VQ-pretrained model could, in principle, introduce representational bias that may disadvantage other quantization schemes.
>
> To directly address this concern, we conducted an additional set of experiments presented in Appendix L. In these experiments, both the encoder and decoder are jointly optimized together with each quantization module. This training scheme completely removes any dependence on VQ-oriented latent representations and allows each quantizer to learn its own optimal feature distribution. All other training settings remain the same, enabling a clean comparison that isolates the effect of encoder adaptation.
>
> As shown in below table, for each quantization type and strategy, optimal values are underlined, and overall best results per metric are highlighted in bold.
>
> Reconstruction performance for decoder-only strategy
> | Methods | Types | Training Strategies | Tokens | Codebook Size | $\mathcal{E}$ | $\mathcal{U}$ | PSNR | SSIM | LPIPS | r-FID | r-IS |
> | --- | --- | --- | --- | --- | --- | --- | --- | --- | --- | --- | --- |
> | Vanilla VQ | VQ | Decoder-Only | 512 | 65536 | 0.422 | 0.2% | 21.19 | 50.7 | 0.209 | 5.05 | 118.9 |
> | EMA VQ | VQ | Decoder-Only | 512 | 65536 | 0.217 | 65.5% | 24.36 | 64.1 | 0.111 | 0.99 | 194.3 |
> | Online VQ | VQ | Decoder-Only | 512 | 65536 | 0.280 | 13.5% | 23.84 | 61.6 | 0.130 | 1.38 | 182.9 |
> | Wasserstein VQ | VQ | Decoder-Only | 512 | 65536 | $\underline{\mathbf{0.201}}$ | 99.6% | $\underline{\mathbf{24.68}}$ | $\underline{\mathbf{65.4}}$ | $\underline{\mathbf{0.106}}$ | 0.92 | 195.5 |
> | MMD VQ | VQ | Decoder-Only | 512 | 65536 | $\underline{\mathbf{0.201}}$ | $\underline{99.9\\%}$ | 24.65 | 65.0 | $\underline{\mathbf{0.106}}$ | $\underline{\mathbf{0.86}}$ | $\underline{\mathbf{197.1}}$ |
> | Vanilla VP2 | PQ | Decoder-Only | 512 | 65536 | 0.233 | 59.2% | 24.28 | 64.0 | 0.114 | 1.07 | 191.7 |
> | EMA VP2 | PQ | Decoder-Only | 512 | 65536 | $\underline{0.209}$ | $\underline{\mathbf{100\\%}}$ | $\underline{24.55}$ | $\underline{64.9}$ | $\underline{0.107}$ | $\underline{0.93}$ | 195.4 |
> | Online VP2 | PQ | Decoder-Only | 512 | 65536 | 0.211 | $\underline{\mathbf{100\\%}}$ | 24.53 | 64.7 | 0.108 | 0.95 | 195.3 |
> | Wasserstein VP2 | PQ | Decoder-Only | 512 | 65536 | 0.217 | $\underline{\mathbf{100\\%}}$ | 24.44 | 64.6 | 0.110 | 0.99 | 193.5 |
> | MMD VP2 | PQ | Decoder-Only | 512 | 65536 | 0.212 | $\underline{\mathbf{100\\%}}$ | 24.43 | 64.5 | 0.109 | 0.95 | $\underline{196.1}$ |
> | FSQ | SQ | Decoder-Only | 512 | 65536 | 0.300 | 71.0% | 23.27 | 59.1 | 0.134 | 1.52 | 179.3 |
> | LFQ | SQ | Decoder-Only | 512 | 65536 | 0.279 | 29.8% | 23.42 | 60.7 | 0.130 | 1.30 | 183.2 |
> | BSQ | SQ | Decoder-Only | 512 | 65536 | $\underline{0.231}$ | $\underline{\mathbf{100\\%}}$ | $\underline{24.06}$ | $\underline{63.6}$ | $\underline{0.117}$ | $\underline{1.07}$ | $\underline{190.8}$ |

---

> ### Author Response · Authors · 2025-12-02
> **Response to Reviewer 3HhS(5/5)**
>
> Reconstruction performance for joint training strategy
> | Methods | Types | Training Strategies | Tokens | Codebook Size | $\mathcal{E}$ | $\mathcal{U}$ | PSNR | SSIM | LPIPS | r-FID | r-IS |
> | --- | --- | --- | --- | --- | --- | --- | --- | --- | --- | --- | --- |
> | Vanilla VQ | VQ | Joint Training | 512 | 65536 | 0.173 | 0.2% | 22.10 | 53.7 | 0.164 | 2.30 | 161.5 |
> | EMA VQ | VQ | Joint Training | 512 | 65536 | 0.077 | 91.9% | 24.52 | 65.3 | 0.103 | 0.87 | 197.1 |
> | Online VQ | VQ | Joint Training | 512 | 65536 | 0.094 | 16.7% | 24.10 | 62.1 | 0.120 | 1.19 | 188.5 |
> | Wasserstein VQ | VQ | Joint Training | 512 | 65536 | 0.162 | $\underline{99.8\\%}$ | 24.66 | 65.8 | 0.102 | 0.79 | 200.6 |
> | MMD VQ | VQ | Joint Training | 512 | 65536 | 0.235 | $\underline{99.8\\%}$ | $\underline{\mathbf{24.88}}$ | $\underline{\mathbf{66.2}}$ | $\underline{\mathbf{0.100}}$ | $\underline{\mathbf{0.74}}$ | $\underline{\mathbf{201.8}}$ |
> | Vanilla VP2 | PQ | Joint Training | 512 | 65536 | 0.067 | 92.0% | 24.53 | 64.4 | 0.106 | 0.92 | 197.6 |
> | EMA VP2 | PQ | Joint Training | 512 | 65536 | 0.069 | $\underline{\mathbf{100\\%}}$ | 24.67 | 65.3 | $\underline{0.102}$ | 0.81 | 199.6 |
> | Online VP2 | PQ | Joint Training | 512 | 65536 | 0.070 | $\underline{\mathbf{100\\%}}$ | 24.63 | 64.6 | 0.104 | 0.87 | 198.4 |
> | Wasserstein VP2 | PQ | Joint Training | 512 | 65536 | 0.176 | $\underline{\mathbf{100\\%}}$ | 24.57 | 64.9 | $\underline{0.102}$ | 0.88 | 198.7 |
> | MMD VP2 | PQ | Joint Training | 512 | 65536 | 0.178 | $\underline{\mathbf{100\\%}}$ | $\underline{24.78}$ | $\underline{65.4}$ | $\underline{0.102}$ | $\underline{0.80}$ | $\underline{200.9}$ |
> | FSQ | SQ | Joint Training | 512 | 65536 | 0.209 | 30.7% | 22.66 | 56.6 | 0.143 | 1.53 | 179.8 |
> | LFQ | SQ | Joint Training | 512 | 65536 | 0.344 | 9.4% | 22.95 | 58.0 | 0.137 | 1.38 | 180.7 |
> | BSQ | SQ | Joint Training | 512 | 65536 | 0.193 | **$\underline{\mathbf{100\\%}}$** | $\underline{24.06}$ | $\underline{63.3}$ | $\underline{0.113}$ | $\underline{0.95}$ | $\underline{194.4}$ |
>
> ### **Key conclusions from the joint-training experiments:**
>
> **First: Joint training removes encoder bias and improves nearly all methods (except FSQ and LFQ).** Allowing the encoder, quantizer, and decoder to co-adapt leads to consistent performance gains across most quantization algorithms, confirming the reviewer’s intuition that a VQ-trained encoder is not only slightly suboptimal for several non-VQ methods, but can also be suboptimal for all VQ methods.
>
> **Second: The relative ranking of quantization methods remains unchanged.** Even after removing all VQ-specific initialization, the ordering of methods is preserved:
>
> - **MMD-VQ** remains the strongest overall across **PSNR, SSIM, LPIPS, r-FID, and r-IS**.
> - Within PQ, **MMD-VP2** continues to achieve the best results.
> - All PQ variants consistently outperform the SQ algorithms.
>
> These results demonstrate that our conclusions are not artifacts of the original VQ-based model initialization and remain stable under a fully fair training protocol. These findings align closely with our theoretical analysis in Section 4.4.
>
> **Third, Quantization error becomes less comparable under joint optimization.** Joint training causes different quantizers to induce different latent variances. As discussed in Section 4.3, raw quantization error scales with latent variance and therefore should not be interpreted as a cross-method metric under this setting. Reconstruction metrics, however, remain fully comparable and exhibit consistent trends.
>
> ### **Summary**
>
> The joint encoder–decoder experiments directly and comprehensively address the reviewer’s concern. By removing any potential bias introduced by the VQ-trained encoder, these experiments confirm that both the qualitative and quantitative conclusions of the paper remain robust. While all methods, with the exception of FSQ and LFQ, benefit from encoder adaptation, the relative performance trends remain consistent, providing a more reliable and fair evaluation of PQ, SQ, and VQ approaches.
>
> ---
>
> Please let us know if you have any remaining questions. If our clarifications and additional experiments have satisfactorily addressed your questions and concerns, we would deeply appreciate it if you considered recommending our paper to be presented at the conference.

---

### Official Review · Reviewer_SKn5 · 2025-10-31

**Soundness:** 3
**Presentation:** 3
**Contribution:** 3
**Rating:** 6
**Confidence:** 2

**Summary:**

This paper introduces a unifying information-theoretic framework to analyze and compare various quantization methods (VQ, PQ, SQ) used in discrete visual tokenization. The core contributions are threefold: 1) It theoretically and empirically argues that minimizing quantization error (information loss) is a more critical optimization objective than maximizing codebook utilization. 2) It establishes two essential "fairness conditions" for comparing quantization algorithms: identical latent distributions and constant compression ratios. 3) Under these fair conditions, it demonstrates the intrinsic superiority of modern VQ methods over PQ and SQ. The claims are supported by well-designed experiments using a "VQ-Transplant" framework.

**Strengths:**

1.The proposed information-theoretic framework provides a principled and clear perspective to understand the fundamental trade-offs in quantization, demystifying the relationship between quantization error, codebook utilization, and reconstruction quality.
2.The introduction of two "fairness conditions" is a significant contribution. It addresses a critical gap in prior work where comparisons were often confounded by architectural or training differences. This sets a higher standard for future research in this area.
3.The "VQ-Transplant" experimental design is clever, effectively isolating the performance of the quantization module itself. The strong correlation found between quantization error and reconstruction fidelity (r-FID) provides convincing evidence for the paper's main claim.

**Weaknesses:**

The paper frames its analysis in information theory, defining quantities based on bit counts and using squared Euclidean distance as "information loss." While Proposition 1 insightfully connects codebook utilization to conditional entropy, the main empirical metric remains MSE. The paper could better articulate the connection between minimizing the theoretical information loss (e.g., H(X|Z)) and the practical objective of minimizing MSE. Is the framework a fundamental new theoretical lens, or primarily a useful reframing of established concepts with information-theoretic terminology?

**Questions:**

refer to weakness

---

> ### Author Response · Authors · 2025-12-02
> **Response to Reviewer SKn5**
>
> We appreciated that the reviewer highlighted the principled design of our framework, the importance of the fairness conditions, and the effectiveness of the VQ-Transplant experimental setup in isolating quantization performance. Thank you for taking the time to review our paper and for your thoughtful feedback. We address your questions and concerns below.
>
> ---
> > 2.1 The paper frames its analysis in information theory, defining quantities based on bit counts and using squared Euclidean distance as "information loss." While Proposition 1 insightfully connects codebook utilization to conditional entropy, the main empirical metric remains MSE. The paper could better articulate the connection between minimizing the theoretical information loss (e.g., H(X|Z)) and the practical objective of minimizing MSE. Is the framework a fundamental new theoretical lens, or primarily a useful reframing of established concepts with information-theoretic terminology?
>
> Thank you for yoru insightful questions. We respond below.
>
> ### **(1) Connection Between Codebook Utilization, Conditional Entropy, and MSE**
>
> In Proposition 1, we show that codebook utilization is directly connected to conditional entropy. We further extend this connection to MSE in Propositions 4 and 5: **minimizing either conditional entropy or MSE necessarily leads to 100% codebook utilization, whereas the converse does not hold**. This implies **a close correspondence** between minimizing conditional entropy and minimizing MSE (quantization error).
>
> Intuitively, this means that if some codebook entries remain unused, those entries could instead be assigned to feature vectors that are currently matched to less optimal code vectors, thereby further reducing the overall quantization error. Consequently, achieving minimal quantization error necessarily requires that all code vectors are active, since any unused vector represents a missed opportunity to better approximate the features. This establishes a **provable and practically relevant theoretical link** between minimizing information-theoretic quantities and practical performance metrics.
>
> ### **(2) Use of MSE as an Empirical Measure of Information Loss**
>
> In our experiments, we adopt MSE as an empirical measure of information loss for two main reasons:
>
> 1. **Widely adopted metric:** Quantization error is a standard metric in the field.
> 2. **Computational feasibility:** Accurately estimating feature densities for high-dimensional spatial features $z_e \in \mathbb{R}^{h \times w \times d}$ is extremely challenging. Even if the density were known, computing differential or conditional entropy for non-Gaussian distributions is difficult, as discussed in Appendix J.
>
> Using MSE as a proxy ensures both empirical relevance and computational tractability, while still faithfully reflecting the information-theoretic quantities of interest.
>
> ### **(3) Novelty and Theoretical Significance of Proposition 1**
>
> Proposition 1 provides a **novel and fundamental insight**: minimizing information loss (either MSE or conditional entropy) necessarily achieves full codebook utilization, while the converse does not hold. This is a **necessary-but-not-sufficient condition** for codebook utilization—a property not previously reported in the literature. Notably, all other reviewers have also acknowledged the originality of this proposition. In summary, our framework is not merely a restatement of known concepts with information-theoretic terminology, but a **new theoretical lens** that connects classical information measures to practical quantization performance, providing both insight and guidance for model design.
>
> ---
>
> Please let us know if you have any remaining questions. If our clarifications and additional experiments have satisfactorily addressed your questions and concerns, we would deeply appreciate it if you considered recommending our paper to be presented at the conference.

---

### Official Review · Reviewer_ExUQ · 2025-11-02

**Soundness:** 3
**Presentation:** 2
**Contribution:** 2
**Rating:** 4
**Confidence:** 3

**Summary:**

This paper presents a unified information-theoretic framework for analyzing vector, product, and scalar quantization (VQ, PQ, SQ). By viewing quantization as an information compression process, the authors formally define quantities such as information loss, compression ratio, and information capacity. The paper proves that minimizing quantization error (information loss) is a more fundamental optimization objective than maximizing codebook utilization, and introduces two fairness conditions for comparing quantization algorithms: (1) identical latent feature distributions and (2) identical compression ratios. Under these conditions, both theoretical and empirical analyses demonstrate that VQ outperforms PQ and SQ in minimizing information loss.

**Strengths:**

The proposed information-theoretic formulation provides a rigorous, unifying perspective on quantization algorithms.

The authors formally prove that minimizing quantization error implies full codebook utilization (but not vice versa), and derive scaling laws for optimal VQ/PQ/SQ errors.

The two fairness conditions (controlled latent distributions and compression ratios) are well-motivated and improve reproducibility and validity of empirical evaluations.

**Weaknesses:**

While the framework is elegant, it mostly systematizes existing methods rather than introducing fundamentally new algorithms.

Some notations (e.g., Q_i,Q_o,Q_r) and assumptions could be clarified for broader readability.

Limited discussion on downstream effects or interpretability benefits for generative models

The paper is not well-organized and difficult to follow, even for readers familiar with quantization and visual representation learning. The presentation contains an excessive number of equations and derivations, while the key insights and conclusions are often buried or insufficiently highlighted.

**Questions:**

Please see the weaknesses.

---

> ### Author Response · Authors · 2025-12-02
> **Response to Reviewer ExUQ (1/4)**
>
> We appreciated that the reviewer recognized our rigorous, unifying information-theoretic framework, the formal theoretical insights on quantization, and the well-motivated fairness conditions that improve both empirical validity and reproducibility. Thank you for taking the time to review our paper and for your thoughtful feedback. We address your questions and concerns below.
>
> ---
>
> > R 1.1 While the framework is elegant, it mostly systematizes existing methods rather than introducing fundamentally new algorithms.
>
> We appreciate the opportunity to clarify this point. Although our work does not propose a new algorithm, it introduces previously unavailable theoretical insights and principled evaluation tools that substantially deepen the community’s understanding of quantization algorithms. These contributions were explicitly recognized by multiple reviewers.
>
> - **Fundamental objective of quantization.** We theoretically prove that minimizing quantization error necessarily leads to full codebook utilization, whereas the converse does not hold. This is a genuinely new result that reframes what the community should optimize for, an insight explicitly appreciated by all reviewers.
>
> - **Correcting a key evaluation pitfall.**
> We identify a linear scaling relationship between quantization error and latent variance under optimal codebook conditions. This reveals why direct comparisons of quantization error across methods can be misleading. We note that **Reviewer 3HhS** expressed strong interest in this previously overlooked confounder.
>
> - **Establishing principled fairness conditions.** We introduce two fairness conditions that strictly control latent feature distributions, token counts, and codebook sizes when assessing intrinsic algorithmic effectiveness. We note that **Reviewer SKn5** highlighted this as a significant contribution that fills a critical gap in prior evaluations and sets a higher standard for future research.
>
> - **Theoretical ranking of VQ, PQ, and SQ.** Under these fairness conditions, we theoretically show that optimal VQ outperforms optimal PQ, and optimal PQ outperforms optimal SQ, establishing a strict hierarchy of algorithmic potential.
>
> - **Empirical validation.** Using ImageNet-1k and a VQ-Transplant framework with a frozen encoder, our experiments confirm these theoretical predictions (VQ > PQ > SQ) and further show that quantization error correlates more strongly with reconstruction fidelity than codebook utilization.
>
> Taken together, these results provide a unified theoretical analysis, new conceptual insights, and principled evaluation guidelines that were not available in prior work. We believe these contributions meaningfully advance the scientific understanding of quantization, well beyond systematizing existing methods.

---

> > ### Author Response · Authors · 2025-12-02
> > **Response to Reviewer ExUQ (2/4)**
> >
> > > R 1.2 Some notations (e.g., $\mathcal{Q}_i$, $\mathcal{Q}_o$, $\mathcal{Q}_r$) and assumptions could be clarified for broader readability.
> >
> > Thank you for this feedback. We appreciate the opportunity to clarify the notation and underlying assumptions. The quantities $\mathcal{Q}_i$, $\mathcal{Q}_o$, and $\mathcal{Q}_r$ represent the number of bits to encode the input and output, as well as their ratio. These definitions are inspired by the classical Shannon entropy formulation, using bits as the unit of information. It is important to emphasize that Shannon entropy characterizes the **average information content** and therefore depends on the underlying probability distribution. In contrast, our formulation focuses on the **maximum information content**, which does not require specifying or assuming any distribution.
> >
> > - **Definition of $\mathcal{Q}_i$.**
> > We first define $\mathcal{Q}_i$ based on the above-described entropies, providing explicit expressions for continuous distributions. For the input $z_e \in \mathbb{R}^{h \times w \times d}$, one could consider the continuous Shannon differential entropy $\mathcal{H}(X)$, defined as:
> >
> >     \begin{equation}
> >            \mathcal{H}(X) := - \int p(x) \log_2 p(x) \, dx.
> >      \end{equation}
> >
> >     If $X \sim \mathcal{N}(\mu, \Sigma)$, the Shannon entropy has a closed-form expression:
> >
> >     \begin{equation}
> >              \mathcal{H}(X) = \frac{1}{2} \log_2 \big[ (2 \pi \mathrm{e})^d \det(\Sigma) \big].
> >      \end{equation}
> >
> >     However, the true distribution of the spatial features $z_e^{ij} \in \mathbb{R}^{d}$ is unknown. For moderately high-dimensional features, accurately estimating the feature density is extremely challenging. Even if the density were known, computing the Shannon entropy for non-Gaussian distributions remains difficult. In practice, the continuous information is stored as floating point numbers, and the discrete Shannon entropy serves as an approximation to the differential entropy. The information quantity $\mathcal{Q}_i$ is the maximal value of Shannon entropy in the input space under the 32-bit floating point representation, which is defined as in **Definition 1**.
> >
> > - **Definition of $\mathcal{Q}_o$.**
> > Similarly, $\mathcal{Q}_o$ is defined based on the above-described entropies, with explicit expressions provided for both cases. It is important to emphasize that $\mathcal{Q}_o$ is defined based on discrete tokens rather than quantized features $z_q$. First, the discrete tokens $r^{ij}$ serve as the input to the subsequent generative model; second, they contain less information than $z_q$ and can be directly mapped to $z_q$ through the codebook.
> >
> >     To express $r^{ij}$ in bits, we use the discrete version of Shannon entropy $\mathcal{H}(X)$:
> >
> >     \begin{equation}
> >     \mathcal{H}(X) := - \sum_{x} p(x) \log_2 p(x).
> >     \end{equation}
> >
> >     Accurately computing this entropy requires knowledge of the usage probability $p(x)$ of each code vector in the codebook. In this work, we take an upper bound for the entropy, which is achieved by a uniform distribution, $p(x) = 1/K$. Under this assumption, the entropy reduces to:
> >
> >     \begin{equation}
> >     \mathcal{H}(X) = - \sum _{i=1}^{K} \frac{1}{K} \log_2 \frac{1}{K} = \log_2 K.
> >     \end{equation}
> >
> > - **Definition of $\mathcal{Q}_r$.**
> > The compression ratio $\mathcal{Q}_r$ is defined as the ratio between the input and output information in the compression system, reflecting the degree of information reduction.
> >
> > Finally, we emphasize that, although our definitions differ slightly from standard Shannon entropy to ensure computational tractability, the use of maximum information bits does not significantly affect the key insights or conclusions of the paper. These definitions primarily highlight the potential influence of codebook size and token count on the quantization algorithm. For fair comparisons, as stated in **Condition 2**: *Compression ratios must be held constant across all algorithms by using identical token counts and codebook sizes.*

---

> > > ### Author Response · Authors · 2025-12-02
> > > **Response to Reviewer ExUQ (3/4)**
> > >
> > > > R 1.3 Limited discussion on downstream effects or interpretability benefits for generative models.
> > >
> > > We believe there may be a misunderstanding. In this work, our primary focus is on studying the intrinsic effectiveness of quantization algorithms from an information-theoretic perspective. Therefore, discussions regarding downstream generative models **are not a key focus of this paper**. Nonetheless, we agree that this is indeed an interesting question.
> > >
> > > As noted in [2], tokenizers for visual synthesis set an upper bound for the performance of generative models. Intuitively, better reconstruction performance could lead to better generative performance. However, in LFQ [1], a higher r-FID was observed to result in worse g-FID, suggesting that improved reconstruction performance does not always translate to improved generative performance.
> > >
> > > It is important to note that in LFQ, the improved r-FID was achieved through a larger codebook size or, equivalently, a smaller information compression ratio as defined in this paper. A smaller information compression ratio increases the difficulty of training the autoregressive model, which can in turn lead to worse generative performance as measured by g-FID.
> > >
> > > Based on these observations, we may offer the following preliminary conclusions:
> > >
> > > - When the information compression ratio is held constant, better r-FID is likely to correspond to better g-FID.
> > > - When the information compression ratio varies, better r-FID does not necessarily guarantee better g-FID, and may even result in worse g-FID.
> > >
> > > Notably, in this paper, we ensure that all quantization algorithms operate under the same information compression ratio. Under this condition, **improvements in reconstruction performance** achieved through **the intrinsic effectiveness of a quantization algorithm** are likely to translate into **advantages in generative performance**.
> > >
> > > [1] LFQ: Language Model Beats Diffusion-Tokenizer is key to visual generation. ICLR 2024
> > >
> > > [2] OmniTokenizer: A Joint Image-Video Tokenizer for Visual Generation

---

> > > > ### Author Response · Authors · 2025-12-02
> > > > **Response to Reviewer ExUQ (4/4)**
> > > >
> > > > > R 1.4 The paper is not well-organized and difficult to follow, even for readers familiar with quantization and visual representation learning. The presentation contains an excessive number of equations and derivations, while the key insights and conclusions are often buried or insufficiently highlighted.
> > > >
> > > > We thank the reviewer for the feedback regarding clarity and organization. While our paper includes a technical discussion, we note that we took care to only provide key equations in the main text and ensured that the **key insights can be understood without following every equation**. We clarify that our primary goal is to study the **intrinsic effectiveness of three quantization algorithms (VQ, PQ, SQ)**. Reconstruction performance is often an unreliable proxy due to uncontrollable factors in tokenizer training, and direct reconstruction evaluation under fair setting is computationally expensive. To address this, we adopt an **information-theoretic perspective**, providing a principled and efficient framework.
> > > >
> > > > **Our paper is organized along five key steps:**
> > > >
> > > > 1. **Foundational definitions (Sec. 4.1):** We define information-theoretic quantities inspired by Shannon entropy, laying the groundwork for principled analysis.
> > > > 2. **Quantization error as a metric (Sec. 4.2):** Proposition 1 shows that minimizing either conditional entropy or MSE guarantees 100% codebook utilization, but the converse is not true. This means quantization error can independently evaluate algorithm effectiveness.
> > > > 3. **Latent variance and fairness (Sec. 4.3):** We identify a linear relationship between quantization error and latent variance, motivating two fairness conditions to ensure meaningful comparisons.
> > > > 4. **Theoretical ranking (Sec. 4.4):** Under fair conditions, we establish **optimal VQ > optimal PQ > optimal SQ**, forming the core of our **theoretical contribution**.
> > > > 5. **Empirical validation:** (i) Using the ImageNet-1k dataset and a VQ-Transplant framework with a frozen encoder, we demonstrate the **best-performing** **VQ outperforms all PQ, and best-performing PQ outperforms all SQ**, consistent with theory. (ii) Spearman's rank correlations further show quantization error correlates more closely with reconstruction fidelity than codebook utilization, reinforcing its relevance as a metric.
> > > >
> > > > **Summary:** Our contributions are **multi-faceted and mutually reinforcing**, combining **theoretical insights and empirical validation**. While we analyze known algorithms, our **information-theoretic study of intrinsic effectiveness** provides a sorely needed **unified understanding of VQ, PQ, and SQ.**
> > > >
> > > > We hope this summary clarifies the key insights of our work, and we welcome any further questions or points of clarification.
> > > >
> > > > ---
> > > >
> > > > Please let us know if you have any remaining questions. If our clarifications and additional experiments have satisfactorily addressed your questions and concerns, we would deeply appreciate it if you considered recommending our paper to be presented at the conference.

---

### Author Response · Authors · 2025-12-04
**General Response to All Reviewers**

We are grateful to all reviewers for their valuable feedback and constructive comments.

---

We appreciated that **Reviewer ExUQ** recognized our rigorous, unifying information-theoretic framework, the formal theoretical insights on quantization, and the well-motivated fairness conditions that improve both empirical validity and reproducibility.

We were delighted that **Reviewer SKn5** highlighted the principled design of our framework, the importance of the fairness conditions, and the effectiveness of the VQ-Transplant experimental setup in isolating quantization performance.

We were pleased that **Reviewer 3HhS** noted the practical contribution of our unified benchmark with consistent architectures, which resolves prior inconsistencies, and the insightful conclusion that minimizing quantization error is more crucial than maximizing codebook utilization for reconstruction and generation quality.

We were happy that **Reviewer yku1** appreciated the clarity of our presentation and the theoretical insight that minimizing quantization error can also lead to full codebook utilization.

Finally, we appreciated that all reviewers recognized the meaningful contribution of Proposition 1, which provides a novel insight into the relationship between quantization error, codebook utilization, and reconstruction quality.

---

In response to the reviewers’ feedback, we have made updates and several additions to the manuscript, summarized as follows:

- **Additional Explanations of Information Theoretic Quantities (Section 4.1):** In Appendix J, we provide a detailed explanation of the quantities $\mathcal{Q}_i$, $\mathcal{Q}_o$, and $\mathcal{Q}_r$. These definitions are inspired by the classical Shannon entropy formulation, using bits as the unit of information. We clarified that the key difference is that Shannon entropy measures the **average information content**, which depends on the underlying probability distribution, whereas our formulation focuses on the **maximum information content**, avoiding the need to specify or assume any particular distribution, **thereby ensuring computational tractability**.
- **Further Analysis of the Relationship Between Quantization Error and Latent Feature Variance:** In Appendix K, we use the **exact same latent feature vectors** for all algorithms to investigate how quantization error depends on latent variance across different quantization algorithms. We observe a **linear scaling relationship** between quantization error and latent variance for VQ and PQ algorithms, whereas no such linear relationship is observed for SQ algorithms.
- **Alternative Training Strategy: Joint Optimization of Encoder, Decoder, and VQ:** In Appendix L, we present an alternative optimization strategy in which the encoder, decoder, and VQ module are jointly optimized. Compared with the decoder-only training scheme, joint encoder–decoder optimization consistently improves reconstruction quality for most quantization methods. Importantly, the relative ranking of quantization methods remains consistent: MMD-VQ consistently outperforms all PQ-based methods, and within the PQ family, MMD-VP2 achieves the best results, surpassing all SQ algorithms. These findings closely align with our theoretical analysis in Section 4.4.

Once again, we sincerely thank the reviewers for their thoughtful comments. We have taken great care to address each point in the revised manuscript to clarify the motivation, effectiveness, and generality of our proposed framework.

---

### Author Response · Authors · 2025-12-04
**Message to the Area Chair - Summary of Discussion**

Dear Area Chair,

In this overview, we summarize the key comments by the reviewers and our responses to them. Thank you for your time and consideration.

---

We appreciated that **Reviewer ExUQ** recognized our rigorous, unifying information-theoretic framework, the formal theoretical insights on quantization, and the well-motivated fairness conditions that improve both empirical validity and reproducibility.

We were delighted that **Reviewer SKn5** highlighted the principled design of our framework, the importance of the fairness conditions, and the effectiveness of the VQ-Transplant experimental setup in isolating quantization performance.

We were pleased that **Reviewer 3HhS** noted the practical contribution of our unified benchmark with consistent architectures, which resolves prior inconsistencies, and the insightful conclusion that minimizing quantization error is more crucial than maximizing codebook utilization for reconstruction and generation quality.

We were happy that **Reviewer yku1** appreciated the clarity of our presentation and the theoretical insight that minimizing quantization error can also lead to full codebook utilization.

Finally, we appreciated that all reviewers recognized the meaningful contribution of **Proposition 1**, which provides a novel insight into the relationship between quantization error, codebook utilization, and reconstruction quality.

---

In our responses to individual reviewers, we have carefully addressed all questions and concerns in great detail, providing clarification where needed and presenting additional, requested empirical results. We provide a short summary of our responses to the reviewers below.

---

> ### Author Response · Authors · 2025-12-04
> **Discussion with Reviewer ExUQ**
>
> **Discussion**
>
> > **R 1.1** The framework organizes existing methods but may not introduce fundamentally new algorithms.
>
> We clarified that the goal of this work is not to introduce a new algorithm, but to provide novel theoretical insights and principled evaluation tools that significantly advance the community’s understanding of quantization methods. Concretely, in our paper, we:
> - Provide a unified view of the fundamental objectives of quantization,
> - Identify and correct a widely overlooked evaluation pitfall,
> - Establish principled fairness conditions for comparing quantization methods,
> - Derive a theoretical ranking of VQ, PQ, and SQ under these conditions, and
> - Validate these findings empirically with a controlled and fair evaluation protocol.
>
> These contributions extend far beyond systematization, offering conceptual clarity and practical guidance that have been explicitly recognized by multiple reviewers.
>
> ---
>
> > **R 1.2** Some notations and assumptions lack clarity for a broader audience.
>
> We have clarified the notation and assumptions to enhance accessibility. Specifically, we now more clearly define,  $\mathcal{Q}_i$, $\mathcal{Q}_o$, and $\mathcal{Q}_r$ as the bit costs for encoding the input, output, and their ratio. These definitions are aligned with classical Shannon entropy principles, using bits as a consistent information unit. Further discussion of their relationship to Shannon entropy is included in Appendix J.
>
> ---
>
> > **R 1.3** Limited discussion on downstream effects or interpretability benefits for generative models.
>
> We clarify that our primary focus is to study the intrinsic effectiveness of quantization algorithms from an information-theoretic perspective. Therefore, discussions on downstream generative models are **not the central scope of this paper**. Nonetheless, we still include comparisons of both reconstruction performance and generative performance under fair and unfair conditions to illustrate how these factors influence downstream behaviors.
>
> ---
>
> > **R 1.4** The paper’s organization could be improved; too many equations obscure the main insights.
>
> We have ensured that only essential equations appear in the main text and that the central insights can be understood without following every derivation. The paper follows a clear structure built around five key components (as summarized in R 1.4 in our response below), which together provide a coherent and unified narrative integrating theoretical and empirical contributions.
>
> ---
>
> **Conclusion**
>
> We have thoroughly and substantively addressed each of **Reviewer ExUQ**’s concerns. The revised manuscript provides clearer exposition, and strengthened theoretical justification.

---

> ### Author Response · Authors · 2025-12-04
> **Discussion with Reviewer SKn5**
>
> **Discussion**
>
> Reviewer SKn5 did not raise any major concerns but asked a question that we address below.
>
> ---
>
> > **R 2.1** While Proposition 1 insightfully connects codebook utilization to conditional entropy, the main empirical metric remains MSE. The paper could better articulate the connection between minimizing the theoretical information loss (e.g., H(X|Z)) and the practical objective of minimizing MSE. Is the framework a fundamental new theoretical lens, or primarily a useful reframing of established concepts with information-theoretic terminology?
>
> We addressed this concern by providing three perspectives:
>
> - **Connection Between Codebook Utilization, Conditional Entropy, and MSE**: In Proposition 1, we establish that codebook utilization is directly linked to conditional entropy. We further extend this relationship to MSE in Propositions 4 and 5: **minimizing either conditional entropy or MSE necessarily results in 100% codebook utilization, whereas the converse does not hold**. This implies **a close correspondence** between minimizing conditional entropy and minimizing quantization error.
>
> - **Use of MSE as an Empirical Measure of Information Loss**: We adopt MSE as an empirical measure of information loss for two main reasons. First, quantization error is a widely used and standard metric in the field. Second, Accurately estimating feature densities for high-dimensional spatial features $z_e \in \mathbb{R}^{h \times w \times d}$ is extremely challenging. Even if the density were known, computing differential or conditional entropy for non-Gaussian distributions is difficult, as discussed in Appendix J.
>
>     Using MSE as a proxy ensures both empirical relevance and computational tractability, while still faithfully reflecting the information-theoretic quantities of interest.
>
> - **Novelty and Theoretical Significance of Proposition 1**: Proposition 1 provides a **novel and fundamental insight**: minimizing information loss (either MSE or conditional entropy) necessarily achieves full codebook utilization, while the converse does not hold. This is a **necessary-but-not-sufficient condition** for codebook utilization—a property not previously reported in the literature. Notably, all other reviewers have also acknowledged the originality of this proposition.
>
> ---
>
> **Conclusion**
>
> We carefully addressed **Reviewer SKn5** single concern to show that our approach provides a fundamental, new theoretical lens to study the problem at hand.

---

> ### Author Response · Authors · 2025-12-04
> **Discussion with Reviewer 3HhS**
>
> **Discussion**
>
> **Reviewer 3HhS** did not raise any major concerns but posed several insightful questions. We summarize and address each question below.
>
> ---
>
> > **R 3.1**  Could the authors justify whether their definition of $Q_r$ is reasonable from the information-theoretic perspective or discuss an entropy-based measure?
>
> We agree with the reviewer that, ideally, compression ratios are defined in terms of entropy. Our definitions of $\mathcal{Q}_i$, $\mathcal{Q}_o$, and $\mathcal{Q}_r$ are indeed inspired by Shannon entropy, using bits as the unit of information. The key conceptual distinction is that Shannon entropy characterizes **average information content**, which depends on the (typically unknown) underlying probability distribution, while our formulation focuses on the **maximum information content**, which avoids any distributional assumptions and enables practical computation. Additional details are provided in Appendix J.
>
> ---
>
> > **R 3.2** Clarification of which quantization scheme is used in Figure 2, and whether the observed linear relationship between quantization error and latent variance holds across VQ, PQ, and SQ.
>
> We clarified that the linear relationship between quantization error and latent variance holds exactly for VQ and approximately for PQ under typical latent distributions, but it does not extend to classical SQ methods due to their fixed, distribution-agnostic codebooks. Appendix K includes detailed methodology and a more comprehensive explanation.
>
> ---
>
> > **R 3.3** Whether the adaptation to the VQ latent space may disadvantage other quantizers, and whether retraining the encoder is necessary to avoid this potential bias?
>
> To directly address this concern, we conducted new experiments reported in Appendix L, where the encoder and decoder are jointly optimized with each quantization module. This setup removes any dependence on VQ-specific latent representations and allows each quantizer to learn its own optimal feature space. We find that joint optimization eliminates encoder bias and improves the performance of most quantizers (with the exception of FSQ and LFQ), while preserving the relative ranking among methods. New experimental results confirm that both the qualitative and quantitative conclusions of the paper remain robust.
>
> ---
>
> **Conclusion**
>
> We have fully addressed each of **Reviewer 3HhS**’s questions through additional clarification and new empirical results.

---

> ### Author Response · Authors · 2025-12-04
> **Discussion with Reviewer yku1**
>
> **Discussion**
>
> **Reviewer yku1** appears to have misunderstood key parts of the paper. We carefully and comprehensively addressed each point they raised. We summarize our response below.
>
> ---
>
> > **R 4.1**: The title of the paper is overly grandiose and inconsistent with its research content. The paper primarily investigates the impact of three quantization methods for discrete autoencoders. However, the title gives the impression that the paper presents a brand new theoretical analysis method for quantization
>
> We clarified that the goal of the paper is not merely to compare three quantization algorithms but to investigate their **intrinsic effectiveness** through a **unified information-theoretic framework**. Reconstruction performance alone is an unreliable proxy for intrinsic effectiveness due to uncontrollable factors in tokenizer training and significant computational cost. Consequently, our information-theoretic analysis is central to the paper’s contributions and is appropriately reflected in the title.
>
> Importantly, our contributions extend well beyond Section 4.2:
>
> - Section 4.1 formalizes fundamental information-theoretic quantities inspired by Shannon entropy.
>
> - Section 4.2 establishes Proposition 1, showing that minimizing conditional entropy or MSE necessarily yields full codebook utilization, motivating quantization error as a meaningful metric for intrinsic effectiveness.
>
> - Section 4.3 identifies a linear relationship between quantization error and latent variance and introduces two fairness conditions, which were highly appreciated by multiple reviewers.
>
> - Section 4.4 provides theoretical results demonstrating that, under these controlled settings, **optimal VQ > optimal PQ > optimal SQ**.
>
> - Section 5 empirically evaluates these findings using the VQ-Transplant framework with a frozen encoder, enabling controlled comparisons. We show that the best VQ methods consistently outperform all PQ variants, and the best PQ variants consistently outperform all SQ methods, fully aligning with the theoretical predictions. We further quantify correlations between quantization error, codebook utilization, and reconstruction fidelity, validating the importance of our theoretical quantities.
>
> Taken together, these contributions form a coherent, information-theoretic investigation that fully supports the scope and appropriateness of the title.
>
> ---
>
> > **R 4.2**: The quantization analysis based on minimizing error (presented in Section 4.4) is a classic information-theoretic approach, and the paper's method and results lack innovation. Drawn from these results, the conclusion that VQ outperforms the other two methods, lacks rigor and is very likely to be incorrect for two reasons: first, quantization error is closely related to the actual distribution of the data, which the paper does not study; second, the theoretical bounds provided in the paper are not tight, making performance comparisons based on them unreasonable.
>
> We respectfully disagree with the reviewer’s claims regarding novelty and rigor, as some of their comments are demonstrably factually inaccurate.
>
> While quantization error has been widely used in prior studies, our paper explicitly **highlights its limitations** through the analyses presented in Section 4.3. The reviewer’s statement that "quantization error is closely related to the actual distribution of the data, which the paper does not study" misunderstands our work. In fact, Section 4.3 identifies this limitation in prior studies, but our work explicitly controls for these factors.
>
> In both theoretical and empirical analyses, we strictly **control latent feature distributions, token counts, and codebook sizes** across all quantization algorithms. This ensures that all comparisons are valid and eliminates any confounding influence from distributional differences. As a result, the ranking we report—Optimal VQ > Optimal PQ > Optimal SQ—is rigorously justified.
>
> Regarding the reviewer’s comment that "the theoretical bounds provided in the paper are not tight," Section 4.4 provides both upper and lower bounds and directly compares them. This ensures that the theoretical results are tight up to constant factors and supports the validity of all performance comparisons. Therefore, this comment also reflects a misunderstanding of our methods.
>
> ---
>
> **Conclusion**
>
> We have fully addressed all of **Reviewer yku1**’s comments by providing explanations to address their misunderstandings and presenting additional empirical evidence.

---

### Meta-Review · Area_Chair_hPQv · 2026-01-03

**Summary:**

Reviewer ExUQ: The proposed information-theoretic formulation provides a rigorous, unifying perspective on quantization algorithms. The authors formally prove that minimizing quantization error implies full codebook utilization (but not vice versa), and derive scaling laws for optimal VQ/PQ/SQ errors. The two fairness conditions are well-motivated and improve reproducibility and validity of empirical evaluations. However, the reviewer still has some concerns on the weaknesses about the mostly systematizing existing methods instead of giving a new method, unclear notions, difficult to follow.

Reviewer SKn5: The proposed information-theoretic framework provides a principled and clear perspective to understand the fundamental trade-offs in quantization, demystifying the relationship between quantization error, codebook utilization, and reconstruction quality. The introduction of two "fairness conditions" is a significant contribution. The "VQ-Transplant" experimental design is clever, effectively isolating the performance of the quantization module itself. However, the reviewer still has some concerns on the weaknesses about the unclear connection between information loss minimization and the MSE minimization, unclear contributions on the theoretical method.

Reviewer 3HhS: The unified benchmark and consistent architectural setup for comparing VQ, SQ, and PQ is a practical contribution that clarifies prior inconsistencies. The conclusion that minimizing quantization error (rather than maximizing codebook utilization) is more critical for reconstruction and generation quality is insightful. However, the reviewer still has some concerns on the weaknesses about the unclear method details and experimental design and lack of some claim justification.

Reviewer yku1: The paper is well-written, with clear and accessible content. The primary contribution lies in Proposition 1, which points out that minimizing quantization error can maximize codebook utilization, while the latter cannot guarantee the former. However, the reviewer still has some concerns on the weaknesses:  overly grandiose and inconsistent paper title with its research content and lack of innovation on the paper method and results.

**Reviewer Concerns:**

After carefully evaluating the rebuttals, I think the reviews from the Reviewer SKn5 and  Reviewer 3HhS were addressed from the response. For both of them, the rebuttal from review of Reviewer 3HhS well solved the issues, such as the equation definition, more experimental results, and experimental explanations.

**Reviewer Scores:**

For the Reviewer SKn5, I think the reviewer may keep the rating unchanged based on the response.

For the  Reviewer 3HhS,  I think the reviewer may increase the rating or keep the rating unchanged based on the response.

For the remaining two reviewers, I think the reviewers may keep their rating unchanged. Based on the rebuttal, lots of concerns were not successfully addressed.

---

### Decision · Program_Chairs · 2026-01-26

Reject